# Learning Superconductivity from Ordered and Disordered Material Structures

**Pin Chen**[1] **Luoxuan Peng**[1] **Rui Jiao**[2,3] **Qing Mo**[1] **Zhen Wang**[1] **Wenbing Huang**[4,5]
**Yang Liu**[2,3] **Yutong Lu**[1*]

[1] National Supercomputer Center in Guangzhou,
School of Computer Science and Engineering, Sun Yat-sen University
[2]Dept. of Comp. Sci. & Tech., Institute for AI, BNRist Center, Tsinghua University
[3]Institute for AIR, Tsinghua University
[4]Gaoling School of Artificial Intelligence, Renmin University of China
[5] Beijing Key Laboratory of Big Data Management and Analysis Methods, Beijing, China

## Abstract

Superconductivity is a fascinating phenomenon observed in certain materials under certain conditions. However, some critical aspects of it, such as the relationship between superconductivity and materials' chemical/structural features, still need to be understood. Recent successes of data-driven approaches in material science strongly inspire researchers to study this relationship with them, but a corresponding dataset is still lacking. Hence, we present a new dataset for data-driven approaches, namely SuperCon3D, containing both 3D crystal structures and experimental superconducting transition temperature ($T_c$) for the first time. Based on SuperCon3D, we propose two deep learning methods for designing high $T_c$ superconductors. The first is SODNet, a novel equivariant graph attention model for screening known structures, which differs from existing models in incorporating both ordered and disordered geometric content. The second is a diffusion generative model DiffCSP-SC for creating new structures, which enables high $T_c$-targeted generation. Extensive experiments demonstrate that both our proposed dataset and models are advantageous for designing new high $T_c$ superconducting candidates.

## 1 Introduction

The pursuit of high-temperature superconductors is driven by their promising applications in efficient energy transmission, advanced electromagnetics, and quantum computing [6, 36], yet their design is hindered by the enigmatic nature of high-T$c$ unconventional superconductivity. Although BCS theory [21] aids in predicting T$c$ for conventional superconductors through first-principles calculations, these methods are computationally demanding and limited to specific materials, necessitating extensive calculations for electron-phonon coupling. Moreover, the intrinsic disorder in many superconductors poses additional challenges for atomic-level design [39]. Such complexities highlight the need for novel approaches in superconductor research and development.

Benefiting from massive public datasets in materials science, data-driven deep learning has been instrumental in predicting material properties [45], synthesizing structures [13], and more. These methods bypass complex physical theories and are crucial in superconductor research, aiding in $T_c$ prediction models for database analysis [12] and inverse design models for novel structures [60], underscoring deep learning's impact on accelerating superconducting material discovery and design. Specially, Graph Neural Networks (GNN) have been extensively applied to model ordered crystals

---

*Yutong Lu are corresponding authors.

38th Conference on Neural Information Processing Systems (NeurIPS 2024) Track on Datasets and Benchmarks.

[61, 16, 12, 64], fewer methods exist for representing disordered crystals [9], despite their prevalence in nature and databases like ICSD, where over 50% of structures are disordered. Therefore, developing methods to represent disordered structures in graphs is vital, especially for superconductivity research where $T_c$ enhancement often involves doping or applying pressure.

Recently, generative model is widely used in Natural Language Processing (NLP), Computer Vision (CV) and natural science. Inspired by non-equilibrium thermodynamics, Diffusion Models (DM) currently produce State-of-the-Art proteins [55], molecules [25] as well as crystals [62, 27]. However, in the field of crystal structure generation, existing models such as CDVAE utilizes the score matching method for atom coordinates, which does not ensure the translation invariance. DiffCSP focuses on crystal structure prediction tasks, which cannot be applied to design novel periodic materials from scratch.

Given the incomplete understanding of superconducting mechanisms, a data-driven approach shows great promise. Constructing a dataset that captures the structure-to-superconductivity relationship is essential for training AI models aimed at designing superconductors. Hence, we introduce SuperCon3D, a new dataset combining crystal structures and the critical temperature $T_c$ from SuperCon and ICSD. Utilizing SuperCon3D, we have developed two deep learning models for superconductor discovery and design. We propose a transformer-based GNN, SODNet, to analyze crystal geometries, including both ordered and disordered structures, potentially screening the entire ICSD. SODNet achieves SE(3)-equivariance through irreducible representation-based vector space features. Additionally, we introduce DiffCSP-SC, a transformer-based equivariant diffusion model for inverse design, capable of generating novel high $T_c$ superconductor candidates.

The main contributions of our work can be summarized as follows:

- A new dataset SuperCon3D containing both ordered-and-disordered crystal structures and experimental superconducting critical temperature is built for the first time.
- We propose two deep learning models to showcase the possible methods for exploring Supercon3D dataset. The experimental result indicate that our proposed models outperform the existing similar methods.
- Based on our proposed models, we present a list of candidate superconductors for future experimental validation. To the best of our knowledge, this is the first report of the candidate superconductors with disordered structures based on GNN methods.

## 2 Related Work

### 2.1 Superconducting Dataset

The SuperCon database encompasses around 33,000 superconductors, providing only their chemical formulas. Jarvis conducted electron-phonon coupling calculations for 1,058 materials, creating a computational database with BCS superconducting properties [12]. However, BCS theory applies mainly to conventional superconductors, and its predicted $T_c$ values require experimental validation. The recent S2S dataset includes 1,685 entries with crystal structures and binary superconducting labels for machine learning-based discovery [31], but it's geared towards classification tasks. Diverging from these approaches, we constructed a dataset comprising crystal structures and experimental $T_c$ values, suited for regression-based deep learning. Additionally, 3DSC [53] is a dataset that includes both $T_c$ and structural information, comprising over 9,150 data entries obtained through elemental matching and manual doping. In contrast, the data in SuperCon3D is entirely derived from experimental observations in databases.

### 2.2 Crystal Modeling

Crystals are typically depicted as periodic graphs with a repeating minimum unit cell in a 3D lattice. While various equivariant GNN models have been developed for ordered crystal structures [61, 64, 11], research on representing disordered crystals is limited. Disorder, as defined by Müller et al. [38], involves varied orientations of atoms in unit cells, categorized into substitutional and positional disorder. MEGNet models disordered sites as elemental embeddings' linear combinations [9], only suitable for substitutional disorder. Our work aims to establish a comprehensive method for representing disordered graphs in crystals.

## 2.3 Generative Models

Drawing on the concepts of non-equilibrium thermodynamics [52], diffusion models create links between data and prior distributions through forward and backward Markov chains [24]. This method has made significant strides in image generation [46, 44]. Leveraging equivariant GNNs, diffusion models efficiently generate samples from invariant distributions, finding applications in conformation generation [51, 63], ab initio molecule design [26], protein generation [33], and more. The adaptation of diffusion models for crystal generation has also gained traction recently [62, 34, 27]. In our research, we enhance diffusion generative modeling by incorporating an attention-based approach, aimed at reverse-engineering novel superconducting structures with a focus on $T_c$ properties.

## 3 Problem Formulation

### 3.1 From Ordered to Disordered Structures

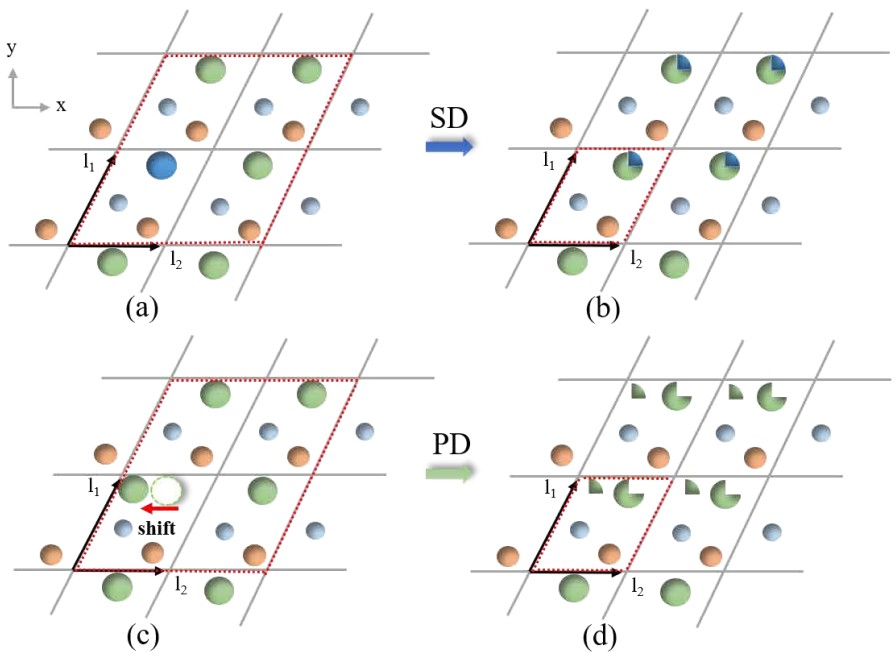

Figure 1: Illustrations of periodic disorder patterns. The dotted red lines are minimum repeated cells. Grey lines are artificial boundaries to form one possible unit cell that repeats in infinite space for the given crystal. (a)→(b): An illustration of periodic substitutional disorder patterns in 2D space. In this case, a new atomic specie replaces the origin one. (c)→(d): An illustration of periodic positional disorder patterns in 2D space. Here, one site occurs position shift, and break the atomic symmetry in the crystal. The crystals are 3D structures in practice, and we use illustrations in 2D for simplicity.

We represent a 3D crystal as the infinite periodic arrangement of atoms in 3D space, and the smallest repeating unit is called a *unit cell*, as shown in Fig. 1. A unit cell can be defined as $\mathcal{M} = (\boldsymbol{L}, \mathcal{S})$, where $\boldsymbol{L} = [\boldsymbol{l}_1, \boldsymbol{l}_2, \boldsymbol{l}_3] \in \mathbb{R}^{3 \times 3}$ represents a minimum unit cell matrix containing three basic vectors to represent the periodicity of the crystal, and $\mathcal{S} = \{S_1, \cdots, S_N\}$ denotes a set of $N$ sites located in the unit cell. Specifically, a *site* describes a composition located at a specific position, which can be further defined as a triplet $S_i = (\boldsymbol{A}_i, \boldsymbol{w}_i, \boldsymbol{x}_i)$, where $\boldsymbol{A}_i = [\boldsymbol{a}_{i,1}, \cdots, \boldsymbol{a}_{i,m_i}] \in \mathbb{R}^{m_i \times h}$ lists the $h$-dimension features of the atom species composing the site, $\boldsymbol{w}_i \in \mathbb{R}^{m_i}$ describes the occupancy of each specie, and $\boldsymbol{x}_i \in \mathbb{R}^3$ denotes the Cartesian coordinate of the site. $m_i$ denotes the number of atoms in one site. Generally a crystal structure is composed of ordered sites, where $m_i = 1$ and $\boldsymbol{w}_i = [1]$, *i.e.* each site is completely formed by a single atom specie. Under the influence of factors such as doping, superconductors may exhibit a disordered structure, containing two kinds of disordered sites:

**Substitutional Disorder (SD)**. As illustrated in Fig. 1(a)→(b), SD involves a situation where the site is occupied by more than one atomic species. Specifically, for an SD site $S_i$, we have

$$\begin{cases} \boldsymbol{m}_i > 1, \\ \boldsymbol{a}_{i,1} \neq \boldsymbol{a}_{i,2} \neq \cdots \neq \boldsymbol{a}_{i,m_i}, \\ \boldsymbol{w}_{i,1} + \boldsymbol{w}_{i,2} + \cdots + \boldsymbol{w}_{i,m_i} = 1 \end{cases} \tag{1}$$

**Positional Disorder (PD)**. In this case, one atom in the unit cell occurs position shift as shown in Fig. 1(c)→(d). For a PD site $S_i$, the atomic specie $\boldsymbol{a}_{i,1}$ partially locates in $\boldsymbol{x}_i$ with its occupancy.

$$\begin{cases} \boldsymbol{m}_i = 1, \\ \boldsymbol{w}_{i,1} < 1. \end{cases} \tag{2}$$

**SD+PD (SPD)**. Specially, when $\boldsymbol{w}_{i,1} + \boldsymbol{w}_{i,2} + \cdots + \boldsymbol{w}_{i,m_i} < 1$ in equation 19, both SD and PD can occur simultaneously.

Typically, there is also the occurrence of interstitial disorder. However, it was not detected in our dataset. Further details are provided in the Appendix B.1.

### 3.2 Superconducting Candidates Designing

In this study, we define the design of novel superconducting candidates in two ways: the first is "known materials repurposing", where the potential superconducting candidates are screened from known structures. And the second involves designing novel material structures that are potential superconducting candidates. The specific definition of the deep learning task is as follows:

**Superconductivity Prediction Task.** The task involves predicting the $T_c$ values given the crystal structure $\mathcal{M}$. Then, we use the predicting models to screen the big structure database to find candidate superconductors with high $T_c$ value.

**Inverse Superconductor Generation Task.** This task predicts the chemical composition $\boldsymbol{A}$, the Cartesian coordinates $\boldsymbol{X}$, and the lattice matrix $\boldsymbol{L}$ targeted on higher $T_c$ values. To reduce the exploration space, we set $\boldsymbol{w}_i = 1$ to generate ordered crystals. Such method can potentially design novel high $T_c$ superconductors.

## 4 The Proposed Method

### 4.1 SODNet

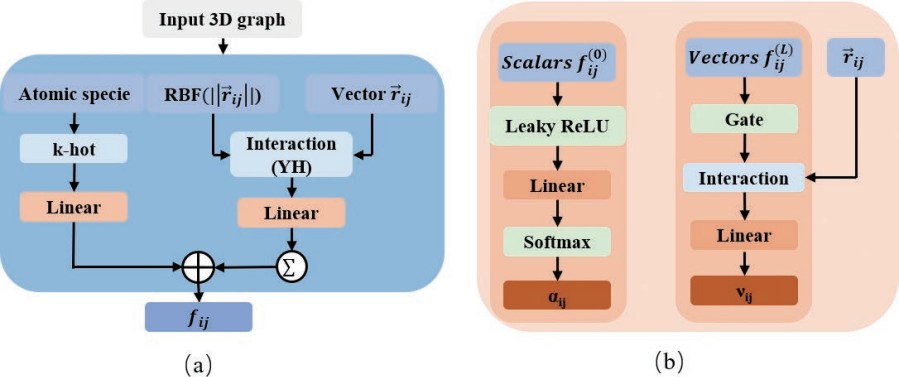

(a)  (b)

Figure 2: Illustration of graph representation and equivariant graph attention layer in SODNet. (a). Illustration of node and edge embeddings. (b). The Type-0 and Type-L features operations in equivariant graph attention mechanism. $\bigoplus$ denotes addition and $\sum$ within a circle stands for summation over all neighbors.

Regarding the importance of symmetry in 3D physics space, it is essential to respect SE(3)-equivariance conditions in neural networks to reduce the model's dependence on data. To explore

the geometric structures with ordered and disordered graphs in SuperCon3D dataset, we propose SODNet, an effective architecture with SE(3)-equivariant graph attention to exploit 3D geometric content. We establish SE(3)-equivariance by utilizing equivariant features derived from vector spaces containing irreducible representations and trainable equivariant operations with the help of e3nn [19]. The core modules of the proposed SODNet is illustrated in Fig. 2. We elaborate the details as follows.

### 4.1.1 Disordered Graph Representation

Considering the presence of disordered structures within SuperCon3D, we design two embedding blocks aimed at enhancing the model's ability to effectively capture these disordered inputs.

**Node embedding**. In the graph network approach, we apply the k-hot embedding [10] as the feature vector $\boldsymbol{a}_{i,k}$, which encodes the atomic property corresponding to each atom specie. To extend such scheme to disordered structures, we further represent each site $S_i$ as a linear combination of atomic occupancy and atomic encoding as:

$$\boldsymbol{h}_i = \begin{cases} \boldsymbol{a}_{i,1}, & S_i \text{ is ordered,} \\ \sum_k \boldsymbol{w}_{i,k}\boldsymbol{a}_{i,k}, & S_i \text{ is SD or SPD,} \\ \boldsymbol{w}_{i,1}\boldsymbol{a}_{i,1}, & S_i \text{ is PD.} \end{cases} \tag{3}$$

**Edge embedding**. Then, we consider 3D geometric features by incorporating interatomic distance as well as vectors $\vec{r}_{ij}$ equipped with spherical harmonics as follows:

$$\boldsymbol{E} = \boldsymbol{w}_i\boldsymbol{w}_j\boldsymbol{RBF}(\|\vec{r}_{ij}\|), \tag{4}$$

$$\boldsymbol{x}_{ij} = \varphi(\boldsymbol{h}_i) + \varphi(\boldsymbol{h}_j), \tag{5}$$

$$\boldsymbol{f}_{ij} = \varphi_f(\boldsymbol{x}_{ij} \otimes_{c\boldsymbol{E}}^{TP} \boldsymbol{SH}(\vec{r}_{ij})) \tag{6}$$

where $\boldsymbol{RBF}(\|\vec{r}_{ij}\|)$ is the radial distribution function (RBF) expansion for interatomic bond distance. Specially, we set $\boldsymbol{w}_i$ and $\boldsymbol{w}_j$ to 1 when $i$ and $j$ sites are ordered. The initial edges are constructed by k-nearest neighbor (kNN) methods from Yan et al. [64]. Here, we remove the close edges when bond distance meets $\|\vec{r}_{ij}\| \le R_i + R_j$ to avoid strong interactions caused by disordered sites, where $R_i$ and $R_j$ are atomic radii. $\boldsymbol{x}_{ij}$ combines the features of target node $i$ and source node $j$ with linear layers to obtain initial message. $\varphi$ represents an MLP. $\boldsymbol{SH}(\vec{r}_{ij})$ is spherical harmonics embeddings (SH) of relative position $\vec{r}_{ij}$, $c\boldsymbol{E}$ is weights parametrized by $\boldsymbol{E}$. Finally, we obtain $\boldsymbol{f}_{ij}$ to derive non-linear messages and attention weights.

### 4.1.2 Equivariant Graph Attention

Given $\boldsymbol{f}_{ij}$ containing multiple type-L vectors, which are SE(3)-equivariant irreps features. In the context of learning on 3D atomistic graphs, it is essential that features and learnable functions exhibit SE(3)-equivariance with respect to geometric transformations acting on the position $\vec{r}_{ij}$. We split $\boldsymbol{f}_{ij}$ into $\boldsymbol{f}_{ij}^L$ and $\boldsymbol{f}_{ij}^0$. The $\boldsymbol{f}_{ij}^0$ is scalar and independent on inputs. However, the $\boldsymbol{f}_{ij}^L$ consists of type-L vectors, which can break equivariance. Inspried by Liao and Smidt [30], we apply different operations to each group of $\boldsymbol{f}_{ij}$.

**Type-0 features.** Given $\boldsymbol{f}_{ij}^0$, we adopt the leaky ReLU activation and a softmax operation for $\beta_{ij}$:

$$\zeta_{ij} = \alpha^\top LeakReLU(\boldsymbol{f}_{ij}^0), \tag{7}$$

$$\beta_{ij} = \frac{exp(\zeta_{ij})}{\sum_{k \in \mathcal{N}(i)} exp(\zeta_{ik})} \tag{8}$$

Where $\alpha$ is a learnable vector of the same dimension as $\boldsymbol{f}_{ij}^0$ and $\zeta_{ij}$ is a scalar.

**Type-L features.** We perform non-linear transformation on $\boldsymbol{f}_{ij}^L$ to obtain non-linear message:

$$\mu_{ij} = Gate(\boldsymbol{f}_{ij}^L), \tag{9}$$

$$\upsilon_{ij} = \varphi_f(\mu_{ij} \otimes_\omega^{TP} \boldsymbol{SH}(\vec{r}_{ij})) \tag{10}$$

We apply the equivariant gate activation as Weiler et al. [59] and present the details in Appendix B.2. Then, the similar method is eq. 6 is used to obtain $v_{ij}$.

Finally, $\beta_{ij}$ and $v_{ij}$ are further transformed features into scalars by multiplication operation. We perform mean aggregate over all nodes to predict the $T_c$ value by:

$$T_c(i) = \frac{1}{|\mathcal{N}(i)|} \sum_{j \in \mathcal{N}(i)} \beta_{ij} \cdot v_{ij}, \tag{11}$$

$$T_c = \frac{1}{|\mathcal{V}|} \sum_{i \in \mathcal{V}} T_c(i) \tag{12}$$

Where $\mathcal{N}(i)$ is the neighbors on node $i$, and $\mathcal{V}$ denotes the set of all nodes in the graph.

### 4.2 DiffCSP-SC

Based on DiffCSP [27], we further equip our method with superconductivity guidance for crystal generation. The original DiffCSP proposes a periodic SE(3) equivariant model to jointly optimize lattice matrix $\boldsymbol{L}$ and fractional coordinates $\boldsymbol{F} = \boldsymbol{L}^{-1}\boldsymbol{X}$ in a diffusion-based framework, and additionally utilizes a time-dependent guidance model [2] for property optimization. Here, $\boldsymbol{X}$ denotes the Cartesian coordinates. We extend DiffCSP with a more powerful architecture for SuperCon3D dataset.

#### 4.2.1 Transformer-based Architecture

The denoising and guidance model of the original DiffCSP share the same architecture, which is built upon EGNN [47], following the standard message passing neural networks (MPNN) framework [20]. To capture the key features related to superconductivity, we employ a transformer-based model for DiffCSP-SC. Let $\boldsymbol{H}^{(s)} = [\boldsymbol{h}_1^{(s)}, \cdots, \boldsymbol{h}_N^{(s)}]$ denote the node representations in the $s$-th layer, where $N$ is the number of nodes. The input feature is given by $\boldsymbol{h}_i^{(0)} = \varphi(f_{\text{atom}}(\boldsymbol{a}_i), f_{\text{pos}}(t))$, where $f_{\text{atom}}$ and $f_{\text{pos}}$ are the atomic embedding and sinusoidal positional encoding [56, 24], respectively. $\varphi$ is a multi-layer perception (MLP).

The output features $\boldsymbol{h}_i^{(s)}$ are computed by

$$\boldsymbol{h}_i^{(s)} = \boldsymbol{h}_i^{(s-1)} + \sum_{j=1}^{N} \theta_{ij}^{(s)} \boldsymbol{v}_{ij}^{(s)} \tag{13}$$

where $\theta_{ij}$ is matrix capturing the similarity between queries and keys.

$$\theta_{ij}^{(s)} = Softmax(\frac{\boldsymbol{q}_i^{(s)\top} \boldsymbol{k}_{ij}^{(s)}}{\sqrt{d}}) \tag{14}$$

Here $d$ is the dimension of the hidden state. The queries, keys and values of $\boldsymbol{q}_i^{(s)}$, $\boldsymbol{k}_{ij}^{(s)}$ and $\boldsymbol{v}_{ij}^{(s)}$ in attention mechanism are unfolded as follows:

$$\boldsymbol{q}_i^{(s)} = \varphi_q(\boldsymbol{h}_i^{(s-1)}), \tag{15}$$

$$\boldsymbol{k}_{ij}^{(s)} = \varphi_k(\boldsymbol{h}_i^{(s-1)}, \boldsymbol{L}^\top \boldsymbol{L}, \psi_{\text{FT}}(\boldsymbol{f}_j - \boldsymbol{f}_i)), \tag{16}$$

$$\boldsymbol{v}_{ij}^{(s)} = \varphi_v(\boldsymbol{h}_i^{(s-1)}, \boldsymbol{L}^\top \boldsymbol{L}, \psi_{\text{FT}}(\boldsymbol{f}_j - \boldsymbol{f}_i)) \tag{17}$$

Where $\varphi_q$, $\varphi_k$ and $\varphi_v$ are MLPs. $\boldsymbol{L}$ is the unit lattice cell. Specially, $\boldsymbol{L}^\top \boldsymbol{L}$ is used to ensure O(3)-equivariance in diffusion step. The transform $\psi_{\text{FT}}$ is able to extract various frequencies of all relative fractional distances that are helpful for crystal structure modeling, and more importantly, $\psi_{\text{FT}}$ is periodic translation invariant, namely, $\psi_{\text{FT}}(w(\boldsymbol{f}_j + \boldsymbol{t}) - w(\boldsymbol{f}_i + \boldsymbol{t})) = \psi_{\text{FT}}(\boldsymbol{f}_j - \boldsymbol{f}_i)$ for any translation $\boldsymbol{t}$. The part corresponding to original DiffCSP is presented in Appendix B.3.

### 4.2.2 Improved Predictor for Evaluation

After denoising process, we need to predict the $T_c$ values of the generated samples. We adopt SODNet as an effective substitute of DFT-based predictors. Similar to CDVAE [62], we calculate the success rate (SR) as the proportion of optimized structures reaching the required thresholds. Given the samples $\tilde{\mathcal{D}}$, SR is defined as

$$\text{SR}\alpha(\tilde{\mathcal{D}}) = \frac{\|\tilde{\mathcal{M}}|\tilde{\mathcal{M}} \in \tilde{\mathcal{D}}, \varphi(\tilde{\mathcal{M}}) > P_{100-\alpha}(\mathcal{D}_{\text{train}})\|}{\|\tilde{\mathcal{D}}\|}, \tag{18}$$

where $\varphi$ is the SODNet predictor and $P_{100-\alpha}(\mathcal{D}_{\text{train}})$ is the $100 - \alpha$ percentile of the $T_c$ values in the training set. Similarly, we define the novelty success rate (NSR) as a metric to assess the generation of novel structures, with detailed definitions and explanations provided in the Appendix D.

### 4.2.3 Pre-training

Considering the SuperCon3D dataset's limited structures, which doesn't fully capture the diversity in atomic species, lattice parameters, and atomic spatial distributions, we pre-trained our model on approximately 1.14 million unique 3D crystals sourced from existing databases, including Materials Project, OQMD, ICSD and Matgen.

## 5 Experiments

### 5.1 Setup

### 5.1.1 SuperCon3D dataset.

We extracted approximately 33,000 superconductors with their chemical formulas and corresponding critical temperatures from SuperCon. After removing duplicates and non-superconductors, we identified 11,949 superconducting materials. Additionally, over 200,000 ordered and disordered crystal structures were gathered from the ICSD database [3]. We then matched these 11,949 SuperCon entries with 208,425 ICSD entries based on chemical composition, space group and lattice parameter. Moreover, $T_c$ values and structural data for hydrogen-enriched superconductors were collated from various literature sources. This process resulted in 1,578 superconductor data entries, each featuring both $T_c$ and crystal structure. To ensure the dataset's integrity, all entries were vetted by domain experts and accompanied by referenced literature. Detailed data descriptions are provided in Appendix A.

### 5.1.2 Evaluation Metrics.

We mainly compare our proposals with other crystal property predictors and inverse crystal structure generative models. For property predicting tasks, we mainly employ Mean Absolute Error (MAE) and R-Square ($R^2$) for $T_c$ prediction. In addition, we also use visualization and interpretable analysis to verify our model. For inverse crystal structure generative task, we calculate the success rate (SR) as the percentage of the 100 optimized structures achieving 10, 30, 50 percentiles of the superconducting property distribution.

### 5.2 Experimental Results and Discussion for Superconductivity Prediction

### 5.2.1 Comparison on Dataset.

We present a summary of comparisons with previous crystal property predictors in Table 1. SODNet consistently outperforms the other competitors both on ordered and disordered structures. For example, SODNet achieves 17.6% reduction on MAE and about 4.4% improvements on $R^2$ than the second ranked Matformer. When considering the PD disordered structure between SODNet and MEGNet, SODNet gets about 41.7% reduction on MAE and almost 66.1% improvement on $R^2$ than MEGNet. It is worth noting that when we incorporate disordered structures into the training and validation sets, the metrics of $R^2$ and MAE both show improvements, indicating that accurately representing disordered structures is beneficial for the prediction of ordered structure properties. This also means that SODNet can be further improved with larger dataset in the future work.

Table 1: Predicting models performance on SuperCon3D dataset. 'O' indicates that using ordered data. ML models with -c and -geo denote composition and structure features.

| Method | Data | | Performance | |
|---|---|---|---|---|
| | Train | Test | MAE (logK)↓ | $R^2$ ↑ |
| RF-c | O | O | 0.738±0.165 | 0.711±0.050 |
| SVM-c | O | O | 0.632±0.094 | 0.801±0.041 |
| RF-geo | O | O | 0.741±0.115 | 0.759±0.051 |
| SVM-geo | O | O | 0.578±0.114 | 0.827±0.042 |
| SchNet | O | O | 0.891±0.041 | 0.401±0.032 |
| CGCNN | O | O | 0.879±0.047 | 0.405±0.022 |
| DimeNet++ | O | O | 0.811±0.058 | 0.434±0.092 |
| SphereNet | O | O | 0.762±0.048 | 0.467±0.096 |
| ALIGNN | O | O | 0.755±0.049 | 0.479±0.090 |
| Matformer | O | O | 0.748±0.043 | 0.570±0.135 |
| MEGNet | O | O | 0.794±0.006 | 0.497±0.009 |
| | O/SD | O/SD | 0.889±0.049 | 0.431±0.058 |
| SODNet | O | O | 0.622±0.112 | 0.595±0.101 |
| | O/SD/PD/SPD | O | **0.584±0.119** | **0.634±0.117** |
| | O/SD | O/SD | **0.518±0.084** | **0.716±0.064** |
| | O/SD/PD/SPD | O/SD/PD/SPD | **0.505±0.055** | **0.748±0.032** |

### 5.2.2 Ablation Study.

We conduct ablation studies to investigate crucial factors that influence the performance of the proposed SODNet. Table 2 shows the experimental results of SODNet with disordered graph representation and equivariant graph attention. When nodes and edges are embedded without atomic occupancy, both the MAE and $R^2$ metrics exhibit a decline in performance. Among them, node embedding is more sensitive to disordered graphs, leading to almost half of the performance loss.

Table 2: Ablation studies of SODNet on SuperCon3D.

| Method | Performance | |
|---|---|---|
| | MAE (logK)↓ | $R^2$↑ |
| *w/o Occupancy Embedding* | | |
| *w/o* disorder node embedding | 0.990±0.033 | 0.365±0.044 |
| *w/o* disorder edge embedding | 0.592±0.087 | 0.655±0.046 |
| *w/o O(3) Equivariance* | | |
| *w/o* equivariant operations | 0.611±0.046 | 0.618±0.027 |
| SODNet | **0.505±0.055** | **0.748±0.032** |

Additionally, if we replace the type-L layer with MLPs, the proposed model achieves worse performance, indicating that the type-L features with equivariant activation function plays a crucial role in O(3) invariance for vectors.

### 5.2.3 Real-world Superconductors Validation

Table 3: Recently discovered superconductors (not included in the training data).

| Material | O/SD/PD | $T_c^{exp}$ (K) | $T_c^{pred}$ (K) | Relative Error(%) |
|---|---|---|---|---|
| $CaH_6$ | O | 215 [35] | 242.25 | 12.67 |
| Ti | O | 26 [66] | 8.50 | 67.31 |
| $CsV_3Sb_5$ | O | 2.3 [18] | 2.36 | 6 |
| $Cs(V_{0.93}Nb_{0.07})_3Sb_5$ | SD | 4.45 [29] | 4.71 | 5.84 |
| $Zr_4Rh_2O$ | O | 3.73 [58] | 4.12 | 10.45 |
| Zr4Pd2O | O | 2.73 [58] | 2.82 | 3.3 |
| $LaFeSiO_{0.9}$ | PD | 10 [23] | 7.93 | 20.7 |

To assess the model's real-world relevance, we gathered newly discovered superconductors from the last three years, not present in our training data. Table 8 reveals that except for titanium superconductors, other materials' critical temperatures ($T_c$) are predicted with low relative error margins (below 21%). This underscores the model's ability to predict $T_c$ values beyond its training scope, highlighting its utility in new material discovery. The outlier predictions for titanium could stem from close atomic proximities under extreme pressures (248 GPa), a condition scarcely represented in our training set. More details are presented in Appendix E.1.

### 5.2.4 Potential Superconducting Materials.

Using our model, we screened the ICSD database to identify potential high-$Tc$ superconductors. Appendix E.2 lists 27 candidates, including cuprate, H-rich, heavy-Fermion, iron-based, and other types. The top three candidates are $Ba_{1.1432}Co_{0.1429}O_{3.0009}Rh_{0.8574}$, $ErH_3$, and $Ba_{0.515}Ca_{0.485}$, previously unreported. This is the first identification of disordered superconducting candidates from ICSD using a GNN method. Given that most ICSD structures are experimentally synthesized, these candidates are valuable for further research. Our model effectively screens disordered high-$T_c$ structures, demonstrating its usefulness. Additionally, we highlight four prime high-$T_c$ candidates with analogous parent structures in Table 10 and 11 of the Appendix E.2. In the Appendix E.3, we provide an interpretation of our SODNet predictor by identifying the features that the model prioritizes when making predictions, using the case of order-and-disorder-$MgB_2$ as an example. This analysis demonstrates SODNet's ability to capture the correlations between superconducting properties and structural characteristics.

## 5.3 Experimental Results and Discussion for Inverse Crystal Structure Generation

### 5.3.1 Comparison on Dataset.

We summarize the comparisons to previous main generative models in Table 4 and present training details in Appendix C.2. Without pretraining, CDVAE, SyMat and DiffCSP generate poor crystal structures, exhibiting extremely low SR performance. The main reason for this phenomenon may be the vast compound space of superconducting materials, making it difficult to effectively sample the atomic species and atomic spatial coordinates. The DiffCSP-SC model we propose shows a slight performance improvement compared to the two models mentioned above under the same conditions. CDVAE lacks translation invariance for atomic coordinates, which affects the quality of generated structures. This low performance metric is also observed in the DiffCSP [27] and aligns with our findings. Moreover, in comparison to DiffCSP, DiffCSP-SC containing an attention mechanism ex-

Table 4: Results for inverse crystal structures generation. "O" and "Pre-training" indicate models trained on SuperCon3D's ordered structures and a collection of 1.14 million stable structures, respectively.

| Model | Data | Performance | | |
|---|---|---|---|---|
| | | SR10 | SR30 | SR50 |
| CDVAE | O | 0.03 | 0.03 | 0.03 |
| SyMat | O | 0.03 | 0.04 | 0.04 |
| DiffCSP | O | 0.04 | 0.05 | 0.05 |
| DiffCSP-SC | O | 0.05 | 0.05 | 0.10 |
| CDVAE | Pre-training + O | 0.25 | 0.25 | 0.30 |
| SyMat | Pre-training + O | 0.28 | 0.28 | 0.35 |
| DiffCSP | Pre-training + O | 0.30 | 0.30 | 0.45 |
| DiffCSP-SC | Pre-training + O | **0.37** | **0.37** | **0.50** |

hibits higher SR performance, indicating that DiffCSP may capture structural features associated with high $T_c$. We will give more discussions for DiffCSP-SC in section *Ablation Study*. Notably, DiffCSP-SC consistently achieved the highest performance across the NSR metric, with detailed results presented in the Appendix D.

### 5.3.2 Ablation Study.

In ablation studies detailed in Table 5, we examine key components of our DiffCSP-SC model. **1.** Assessing the transformer's impact, its removal and reverting to the original DiffCSP approach led to a notable performance drop, especially in SR10 and SR30 metrics, implicating a decrease in high $T_c$ superconductor generation. This suggests that attention mechanisms in transformers effectively capture the complex atomic compositions of high $T_c$ superconductors, which often involve multi-component, multi-element structures. **2.** The pre-training methodology's significance is highlighted by its ability to manage the vast feature space of atomic species, coordinates, and unit cells in ordered crystals. Without it, as seen when training solely on a limited subset from SuperCon3D, the model's efficacy in generating valid superconductors significantly diminishes.

### 5.3.3 Candidate Superconductors.

Utilizing our model, we aimed to generate novel superconductor candidates with high $T_c$ values. Table 12 and 13 in Appendix E.4 displays 32 potential high $T_c$ superconducting materials categorized as cuprate, H-rich, heavy-Fermion, iron-based, and other types. We initially assessed the novelty of these structures through similarity calculations with our 1.14 million-structure database. Interestingly, our findings reveal three candidates, index 8, 9, and 12, previously reported for $T_c$ using computational methods. Additionally, another candidate, index 21 and 22, demonstrated superconductivity upon doping and pressing. Subsequently, density-functional theory (DFT) were performed on selected candidates to verify their superconducting properties. Notably, Van Hove singularities (VHS) were observed in the electronic structures of $Ba_2CuCl_2O_2$, Lu, and $BaFe_2Se_2$, as further detailed in Appendix E.5. VHS is a significant aspect in superconductivity research, often explored for its potential influence [5].

Table 5: Ablation studies of DiffCSP-SC on SuperCon3D.

| Method | Performance | | |
| --- | --- | --- | --- |
| | SR10 | SR30 | SR50 |
| $w/o\,Transformer$ | | | |
| $w/o$ attention | 0.28 | 0.28 | 0.45 |
| $w/o\,Pre\text{-}training$ | | | |
| $w/o$ pre-training | 0.05 | 0.05 | 0.10 |
| DiffCSP-SC | **0.37** | **0.37** | **0.50** |

## 6 Conclusion and Discussion

In conclusion, a novel dataset has been constructed as a benchmark for future deep learning-based superconductivity research. Utilizing the dataset, we put forth two deep learning approaches for the design of high $T_c$ superconductors: a property prediction model for screening the known structures, and a generative model for creating the novel structures. To further validate the efficacy of the model, we apply the predicting model to screen the entire ICSD and identify a list of ordered and disorder superconducting candidates. By employing pretraining on large-scale crystal structures, we have achieved the capability to perform reverse structure design on limited superconducting data points.

Our SuperCon3D dataset, featuring experimental structures and $T_c$ values, paves the way for real-world superconductor applications. Combined with SODNet, which addresses disordered graph issues previously overlooked by the AI community, and DiffCSP-SC for novel designs. However, the accuracy of data-driven models remains constrained by the collected superconducting dataset. As Fig. 4 in the Appendix shows, data unevenness and elemental skewness (especially in Cu and O) may bias the model. Additionally, as Table 8 indicates, atomic distributions under extreme pressures contribute to predictive errors. Addressing these, Fig. 8 presents our pipeline, combining DiffCSP-SC and SODNet, to design and validate novel superconductors through wet experiments, iteratively enriching the dataset for improved model training and accuracy.

## 7 Acknowledgement

This work was jointly supported by the following projects: National Science and Technology Major Project (2022ZD0117805), the National Natural Science Foundation of China (No. 61925601, No. 62376276), Beijing Nova Program (20230484278).

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

## A    SuperCon3D Data Details

We extracted approximately 33,000 superconductors, including their chemical formulas and critical temperatures, from the SuperCon database[2]. After eliminating duplicates and non-superconductors, we retained 11,949 superconducting materials. Over 200,000 ordered and disordered crystal structures were collected from the ICSD database. We then matched the 11,949 SuperCon entries with 208,425 ICSD entries based on chemical composition, space group and lattice parameter. Specifically, we first performed an initial matching based on chemical composition, which may result in one-to-one or one-to-many matches. We then further refined the matches using additional information provided in the literature, such as space groups and lattice constants. Additionally, $T_c$ values and structural data for hydrogen-enriched superconductors were obtained from literature sources. Ultimately, we compiled 1,578 superconductors with both $T_c$ and crystal structure information.

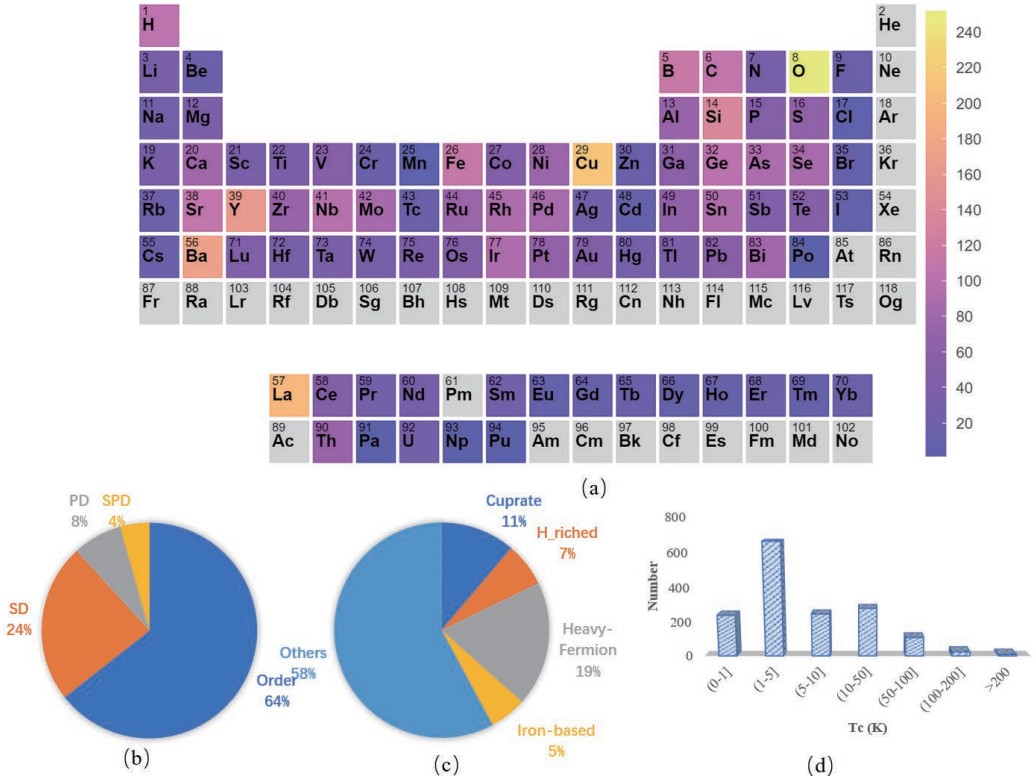

Figure 4: The data distribution of SuperCon3D dataset. (a). The probability of crystals containing a given element in the dataset. (b). The distribution of ordered and disordered superconductors. (c). The distribution of superconducting types. (d). The distribution of $T_c$ values.

We plot the data distribution of SuperCon3D dataset in Fig. 4. In dataset, there are 83 different elements, which encompass most of elemental types found in the periodic table. The most frequent elements are O, Cu, La, Ba, Y as shown in Fig. 4a. Fig. 4b depicts the order and disorder distribution. We classify superconducting materials according to cuprate, H riched, heavy fermion, iron based,

---

[2]https://github.com/vstanev1/Supercon

and others, and distribute the types in Fig. 4c. The distribution of the $T_c$ values of superconducting materials is shown in Fig. 4d. The SuperCon3D dataset can be obtained from the source code package, and the access address is proveide in Sec. G.

## B  Methods

### B.1  Interstitial Disorder

**Interstitial disorder (ID)**. ID refers to the presence of atoms occupying interstitial sites within a crystal lattice, which are not part of the regular lattice positions. These interstitial atoms introduce additional disorder into the structure. The total occupancy, including both regular lattice sites and interstitial sites, can be expressed as:

$$\boldsymbol{w}_{i,1} + \boldsymbol{w}_{i,2} + \cdots + \boldsymbol{w}_{i,m_i} + \boldsymbol{w}_{i,interstitial} = 1 + \Delta \tag{19}$$

where $\boldsymbol{w}_{i,m_i}$ represents the occupancy weight of $m_i$ at site $i$, $\boldsymbol{w}_{i,interstitial}$ represents the occupancy weight of interstitial atoms at site $i$, $\Delta$ is the excess occupancy due to interstitial atoms, with $\Delta > 0$ indicating the presence of ID. In this case, our disordered graph encoding method remains effective.

ID mixed with substitutional disorder (SD) and positional disorder (PD), would result in more new types. However, given the lack of observation of ID in the SuperCon3D dataset, we will not elaborate further on it.

### B.2  Gate layer

We employ the gate activation mechanism [59] for the equivariant activation function. Standard activation functions are applied to type-0 vectors. For higher order vectors (L >0), we achieve equivariance by multiplying them with non-linearly transformed type-0 vectors. Specifically, for an input $x$ comprising non-scalar $C_L$ type-L vectors (where $0 < L \leq L_{\max}$) and $(C_0 + P_L \sum_{L=1}^{L_{\max}} C_L)$ type-0 vectors, we apply SiLU [14] to the first $C_0$ type-0 vectors and a sigmoid function to the remaining $P_L \sum_{L=1}^{L_{\max}} C_L$ type-0 vectors. This process generates non-linear weights, which are then used to scale each type-L vector. After gate activation, the number of channels for type-0 vectors is reduced to $C_0$.

### B.3  The Denoising Method of DiffCSP

We introduce the denoising model $\phi(\boldsymbol{L}, \boldsymbol{F}, \boldsymbol{A}, t)$ as part of the original DiffCSP model, which is related to the *Transformer-based Architecture* section in the main text.

Node representations in the $s$-th layer, $\boldsymbol{H}^{(s)} = [\boldsymbol{h}_1^{(s)}, \cdots, \boldsymbol{h}_N^{(s)}]$, are initialized as $\boldsymbol{h}_i^{(0)} = \psi(f_{\text{atom}}(\boldsymbol{a}_i), f_{\text{pos}}(t))$, combining atomic embeddings $f_{\text{atom}}$ and sinusoidal positional encoding $f_{\text{pos}}$ [24, 56], processed by MLP $\psi$.

Incorporating EGNN [48], the message-passing in layer $s$ is:

$$\boldsymbol{m}_{ij}^{(s)} = \varphi_m(\boldsymbol{h}_i^{(s-1)}, \boldsymbol{h}_j^{(s-1)}, \boldsymbol{L}^\top \boldsymbol{L}, \psi_{\text{FT}}(\boldsymbol{f}_j - \boldsymbol{f}_i)), \tag{20}$$

$$\boldsymbol{m}_i^{(s)} = \sum_{j=1}^N \boldsymbol{m}_{ij}^{(s)}, \tag{21}$$

$$\boldsymbol{h}_i^{(s)} = \boldsymbol{h}_i^{(s-1)} + \varphi_h(\boldsymbol{h}_i^{(s-1)}, \boldsymbol{m}_i^{(s)}). \tag{22}$$

Here, $\varphi_m$ and $\varphi_h$ are MLPs. $\psi_{\text{FT}}$ executes Fourier Transformation on relative fractional coordinates, ensuring periodic translation invariance.

Following $S$ message-passing layers, lattice noise $\hat{\epsilon}_{\boldsymbol{L}}$ is computed as follows:

$$\hat{\epsilon}\boldsymbol{L} = \boldsymbol{L}\varphi_L\left(\frac{1}{N}\sum i = 1^N \boldsymbol{h}_i^{(S)}\right), \tag{23}$$

with $\varphi_L$ shaping output as $3 \times 3$. For fractional coordinate score $\hat{\epsilon}_F$, we have:

$$\hat{\epsilon}_F[:, i] = \varphi_F(\boldsymbol{h}_i^{(S)}), \tag{24}$$

where $\hat{\epsilon}_F[:, i]$ is the $i$-th column, and $\varphi_F$ operates on the final layer's output.

The inner product $\boldsymbol{L}^\top \boldsymbol{L}$ in Eq.(20) ensures O(3)-invariance, as $(\boldsymbol{QL})^\top(\boldsymbol{QL}) = \boldsymbol{L}^\top \boldsymbol{L}$ for any orthogonal $\boldsymbol{Q} \in \mathbb{R}^{3 \times 3}$. This guarantees the O(3)-invariance of $\varphi_L$ in Eq.(24), and $\boldsymbol{L}$ left-multiplied with $\varphi_L$ ensures O(3)-equivariance of $\hat{\epsilon}_L$. Thus, $\phi(\boldsymbol{L}, \boldsymbol{F}, \boldsymbol{A}, t)$ satisfies the proposed properties. More details are described in Jiao et al. [27].

## C  Hyper-parameters and Training Details

In this section, we provide the training details of property predicting models and generative models.

### C.1  Property Predicting Models

We employ the codebase from RF and SVM [53][3], SchNet [50][4], CGCNN [61][5], DimNet++ [16][6], SphereNet [32][7],ALIGNN [11][8], Matformer [65][9] and MEGNet [8][10] for baseline implementations. All models are conducted 10-fold experiments based data splited method of 8:1:1. The training details of each model are as follows:

#### C.1.1  RF and SVM.

We conducted experiments comparing non-deep learning methods, specifically RF and SVM, using both chemical composition features and combined geometric structure features. The SVM model is configured with the following parameters: kernel is set to 'rbf' for mapping data into a higher-dimensional space, while degree, set to 3, controls the complexity of the polynomial kernel (applicable only when a polynomial kernel is used). gamma, set to 'scale', adjusts the influence of individual data points in the feature space. The RF model uses these parameters: n_estimators is set to 100, defining the number of submodels in the ensemble. criterion, set to 'squared_error', evaluates split quality based on mean squared error. min_samples_split, set to 2, specifies the minimum number of samples required to split an internal node, and min_samples_leaf, set to 1, defines the minimum samples needed at a leaf node. Lastly, max_features, set to 1.0, determines the proportion of features considered when finding the best split.

#### C.1.2  SchNet.

Employing the SchNet framework, our method integrates six 64-dimensional message passing layers. SchNet was trained over 500 epochs, using a 5e-4 learning rate and 64 batch size. We optimized using Adam with 1e-5 weight decay, and a one-cycle learning rate scheduler. Atomic radii were determined by the 12th smallest distance between an atom and its neighbors.

#### C.1.3  CGCNN.

A batch size of 64 is employed, and the model consists of three layers of CGCNN message passing layer with 128 hidden dimensions. The training process utilizes the Adam optimizer. Initially, a learning rate of 1e-3 is set for the 200 epochs. During the training, a radius cutoff of 8.0 is applied to all crystals, and the 32 nearest neighbors are selected.

---

[3]https://github.com/aimat-lab/3DSC
[4]https://github.com/atomistic-machine-learning/SchNet
[5]https://github.com/txie-93/cgcnn
[6]https://github.com/gasteigerjo/dimenet
[7]https://github.com/divelab/DIG
[8]https://github.com/usnistgov/alignn
[9]https://github.com/YKQ98/Matformer
[10]https://github.com/materialsvirtuallab/megnet

### C.1.4  DimNet++.

In our approach, we apply a radius cutoff of 8.0 to all crystals and select the 12 nearest neighbors. To represent each node, we utilize Gaussian radial basis function (RBF) kernels. This results in a 64-dimensional embedding for each node. To optimize the model, we employ the Adam optimizer with a weight decay of 1e-6. The model is trained for 500 epochs using a batch size of 128.

### C.1.5  SphereNet.

In our method, we utilize multi-graph representations of materials as inputs to SphereNet models. The input embedding size is set to 256, and the output embedding size is set to 64 for both the 8 LB2 and LB blocks. A cutoff distance of 6 is used. For each model, we initially perform a warm-up on the learning rate, starting at 1e-3. Subsequently, two learning rate strategies—ReduceLROnPlateau and StepLR—are employed for training. In the StepLR strategy, the learning rate is decayed by a specified ratio every fixed number of epochs, known as the step size. The batch size is set to 32, and training is conducted for 300 epochs.

### C.1.6  ALIGNN.

ALIGNN is trained for 150 epochs with a learning rate of 5e-4 and a batch size of 64. The model architecture follows the original paper, consisting of four GCN layers and four ALIGNN layers. The atom feature dimension is set to 92, and the edge feature dimension is set to 80. The training process utilize the Adam optimizer with a weight decay of 1e-5. Additionally, a one-cycle learning rate scheduler is employed. For all crystals, a radius cutoff of 8.0 is applied, and the nearest 12 neighbors are selected.

### C.1.7  Matformer.

In constructing the crystal graph, we follow a specific procedure. The radius for the neighborhood of a given atom is determined by the 12-th smallest distance between that atom and its neighboring atoms. All atoms within this radius are considered part of the neighborhood for the given atom. Each node is then represented by mapping its atomic number to a 92-dimensional embedding using the CGCNN atomic embedding. This embedding is further transformed into a 128-dimensional vector through a linear transformation. Similarly, for each edge, we utilize a 128-dimensional embedding mapping of the Euclidean distance. This mapping is achieved by employing 128 radial basis function (RBF) kernels with centers ranging from 0.0 to 8.0. During the training process, we employ the Adam optimizer with a weight decay of 1e-5. Additionally, a one-cycle learning rate scheduler is utilized. A batch size of 64 is employed and trained for 150 epochs.

### C.1.8  MEGNet.

To construct the crystal graph, we employ three layers of the MEGNET message passing with with 64,32,16 hidden units, and utilize the Set2Set readout function. Following the configuration described in the original paper, MEGNET is trained for 200 epochs using a batch size of 64 and a learning rate of 1e-3. The Adam optimizer with a weight decay of 1e-5 is used for optimization, and a one-cycle learning rate scheduler is implemented. A radius of 8.0 is set for all crystals.

Table 7: Hyper-parameters for SODNet.

| Hyper-parameters | Value or description |
| --- | --- |
| Batch size | 32, 64, 128 |
| Number of epochs | 150, 300 |
| Number of attention heads | 4, 8 |
| Dropout rate | 0.0, 0.1, 0.2 |
| Cutoff radius (Å) | 8, 12, 16 |
| Number of radial bases | 128 |
| Number of transformer blocks | 6 |
| Weigh decay | $0.5 \times 10^{-3}$, $1 \times 10^{-3}$ |

Table 8: Recently discovered superconductors (not included in the training data).

| No. | Material | Type | $T_c^{exp}$ (K) |
|-----|----------|------|------|
| 1 | $CaH_6$ @172 GPa | Order | 215 [35] |
| 2 | Ti @248 GPa | Order | 26 [66] |
| 3 | $CsV_3Sb_5$ | Order | 2.3 [18] |
| 4 | $Cs(V_{0.93}Nb_{0.07})_3Sb_5$ | SD | 4.45 [29] |
| 5 | $Zr_4Rh_2O$ | Order | 3.73 [58] |
| 6 | Zr4Pd2O | Order | 2.73 [58] |
| 7 | $LaFeSiO_{0.9}$ | PD | 10 [23] |

### C.1.9 SODNet

During training, we use a batch size of 64 and trained the model for 150 epochs. A radius of 8.0 is applied to define the neighborhood of each crystal. We utilize 128 basis functions to capture the features of the crystals. To control overfitting, a weight decay of 5e-3 is applied. The learning rate is set to 5e-5, with a minimum learning rate of 1e-6. We employ the AdamW optimizer for efficient optimization. The model architecture consisted of 6 Transformer blocks, each with 8 attention heads. This allowed the model to effectively capture the relationships and dependencies within the crystal structures. Irreps features consist of channels of vectors with degrees up to $L_{\max}$. We denote $C_L$ type-$L$ vectors as $(C_L, L)$ and $C_{(L,p)}$ type-$(L, p)$ vectors as $(C_{(L,p)}, L, p)$. Brackets denote concatenations of vectors. we set irreps features containing 512 type-0 vectors and 128 type-1 vectors, which can be expressed as $[(512, 0), (128, 1)]$. Table 7 summarizes the hyper-parameters for the model.

### C.2 Generative Models

We apply the codebases from CDVAE [62][11], SyMat [34][12] and DiffCSP [27][13] for baseline implementations. All models are conducted experiments based data splited method of 6:2:2. For pretraining, we obtain crystal structures from the databases of Materials Project[14], Open Quantum Materials Database[15], Matgen[16], and ICSD[17]. Molecular crystals are excluded from the dataset. Subsequently, we perform deduplication on all crystal structures, resulting in approximately 1.14 million unique structures. The training specifics for each model are outlined below:

### C.2.1 CDVAE.

For CDVAE model, We replaced the original DimNet++ [16] with SODNet to ensure a fair comparison with other generation models. Regarding the decoder, we utilize the GemNet-T [17], which consists of 3 layers and 128 hidden states.

### C.2.2 SyMat.

For the SyMat model, the property predictor employs SphereNet, which consists of four message-passing layers with a hidden size of 128. The VAE decoder utilizes MLP models composed of two linear layers with a ReLU activation function between them and a hidden size of 256. During training, we use a learning rate of 0.001, a batch size of 128, and run for 1,000 epochs. We assign different weights to various loss terms: 1.0 for atom type set size, 30.0 for atom types, 1.0 for the number of each atom type, and 10.0 for lattice items. Additionally, we apply a weight of 0.01 for the KL-divergence loss and 10.0 for the denoising score matching loss.

---

[11] https://github.com/txie-93/cdvae

[12] https://github.com/divelab/AIRS

[13] https://github.com/jiaor17/DiffCSP

[14] https://next-gen.materialsproject.org

[15] https://www.oqmd.org

[16] https://matgen.nscc-gz.cn

[17] https://icsd.products.fiz-karlsruhe.de/

### C.2.3 DiffCSP.

We employ a configuration of 6 layers with 512 hidden states for datasets other than specified ones. The dimension of the Fourier embedding is set to 256. To control the variance of the DDPM (Diffusion-Driven Probabilistic Modeling) process on $L_t$, we utilize the cosine scheduler with 0.008. Additionally, we use an exponential scheduler with $\sigma_1 = 0.005, \sigma_T = 0.5$ to control the noise scale of the score matching process on $F_t$. The diffusion step is set to 1000. Our model is trained for 1000 epochs, employing the same optimizer and learning rate scheduler as CDVAE.

### C.2.4 DiffCSP-SC.

We utilize SODNet as the property predictor, and the parameter configuration aligns with Table 7. The parameters for the diffusion process also follow the original DiffCSP setup. The difference lies in the message passing layer, where we employ a transformer. Specifically, we use a 512-dimensional hidden state encoding and set the number of heads to 8.

### C.3 Pre-training Dataset

we pre-trained our model on approximately 1.14 million unique 3D crystals sourced from existing databases, including Materials Project, OQMD, Matgen and ICSD.

## D Novel Material Structure Generation

we have introduced a new metric, the "novelty success rate" (NSR), to specifically quantify the proportion of novel structures generated by the model. The NSR is defined as:

$$NSR_\alpha(\tilde{D}) = \frac{\left\| \tilde{M} \mid \tilde{M} \in \tilde{D}, \varphi(\tilde{M}) > P_{100-\alpha}(D_{\text{train}}), \tilde{M} \notin D_{\text{train}} \right\|}{\|\tilde{D}\|} \tag{25}$$

This metric focuses on evaluating the model's ability to generate structures that are not present in the training dataset.

We conducted additional experiments using NSR, and the results are summarized in the table below, comparing different models:

Table 9: NSR comparison across different models and data settings

| Model | Data | NSR10 | NSR30 | NSR50 |
|---|---|---|---|---|
| CDVAE | O | 0.02 | 0.02 | 0.02 |
| SyMat | O | 0.02 | 0.03 | 0.03 |
| DiffCSP | O | 0.03 | 0.04 | 0.04 |
| DiffCSP-SC | O | 0.04 | 0.04 | 0.09 |
| CDVAE | Pre-training + O | 0.19 | 0.19 | 0.25 |
| SyMat | Pre-training + O | 0.20 | 0.21 | 0.26 |
| DiffCSP | Pre-training + O | 0.25 | 0.25 | 0.33 |
| DiffCSP-SC | Pre-training + O | 0.31 | 0.31 | 0.39 |

As shown in table 9, our DiffCSP-SC model outperforms others in generating novel materials, as indicated by higher NSR values across all metrics (NSR10, NSR30, NSR50).

Furthermore, our training dataset includes approximately 1 million material structures, many of which have not been experimentally validated for superconductivity. Even if some generated structures appear in the training data, they may still hold potential superconducting properties, making them valuable for further investigation. By leveraging this large dataset and pre-training strategies, our model demonstrates advantages in generating novel and potentially superconductive structures.

Table 10: The predicted potential candidates of high-$T_c$ cuprate and h-riched superconductors. Candidates of high confidence are marked in gray.

| Type | ICSD code | Chemical formula | O/SD/PD | $T_c$ (K) | Reported SC. |
|---|---|---|---|---|---|
| Cuprate | 68675 | $CuO_2Sr_{0.075}$ | PD | 93.42 | $CuO_2Sr$ 91K [54] |
| | 50774 | $Ca_{0.779}CuO_2Y_{0.041}$ | PD | 65.70 | |
| | 50773 | $Ca_{0.82}CuO_2$ | PD | 64.72 | $CaCuO_2$ 89K [49] |
| | 68217 | $Ba_2CuO_3$ | O | 59.89 | $Ba_2CuO_{3.2}$ 70K [28] |
| | 67394 | $Ba_2CuIO_2$ | O | 43.80 | - |
| H-riched | 187375 | $ErH_3$ | O | 193.03 | - |
| | 635802 | $GdH_3$ | O | 143.19 | - |
| | 623739 | $H_{2.57}Co_{0.14}U_{0.84}$ | PD | 136.76 | - |
| | 42009 | $TbH_{2.25}$ | SD | 135.13 | - |
| | 424154 | $H_6Mg_{1.02}Ti_{1.98}$ | O | 134.34 | - |
| | 230140 | $Li_{0.14}Y_{0.86}H_{2.7}$ | PD | 125.94 | - |
| | 93250 | $YFe_2H_5$ | PD | 125.00 | - |

# E    Potential Superconductors

In this section, we initially validate our model using the $T_c$ values of superconducting materials reported in recent literature, noting that these data points are not included in the SuperCon3D dataset. Subsequently, we present the potential superconducting materials using property prediction model based on SODNet and generative model based on DiffCSP-SC, respectively.

Table 11: The predicted potential candidates of high-$T_c$ heavy-fermion, iron-based and others superconductors. Candidates of high confidence are marked in gray.

| Type | ICSD code | Chemical formula | O/SD/PD | $T_c$ (K) | Reported SC. |
|---|---|---|---|---|---|
| Heavy-Fermion | 168466 | $LaMg_{12}$ | O | 23.83 | - |
| | 161141 | $LaMg_{11.196}Al_{0.804}$ | SD | 21.13 | - |
| | 69897 | $C_2Ce_{0.75}U_{0.25}$ | PD | 11.88 | - |
| | 647197 | $Np_{1.1}Pu_{0.9}$ | SD | 11.75 | - |
| | 614236 | $TmFe_4B$ | O | 10.81 | - |
| Iron-based | 427163 | $Ba_{0.83}Fe_2Rb_{0.17}As_2$ | SD | 23.21 | $Ba_{0.6}Fe_2Rb_{0.4}As_2$ 37.5K [42] |
| | 188347 | $BaFe_2As_2$ | O | 23.27 | - |
| | 39530 | $FeCl_7Te$ | O | 19.57 | - |
| | 633401 | $FeSb_{0.4}Te_{1.6}$ | SD | 16.83 | - |
| | 165523 | $As_2Ba_{0.777}Fe_2$-$K_{0.126}Sn_{0.096}$ | PD | 15.55 | - |
| Others | 96031 | $Ba_{1.1432}Co_{0.1429}$-$O_{3.0009}Rh_{0.8574}$ | PD | 202.12 | - |
| | 58639 | $Ba_{0.515}Ca_{0.485}$ | SD | 160.95 | - |
| | 616160 | $BaSr$ | SD | 123.51 | - |
| | 106111 | $SrTl_2$ | O | 63.52 | - |
| | 428028 | $Ge_{0.6}Sb_{0.27}Te$ | SD | 47.48 | - |

## E.1    Real-world Superconductors Validation

As shown in Table 8, we have collected the structures of superconducting materials along with their corresponding $T_c$ values, as reported in the latest literature over the past three years. This includes a total of seven superconducting materials with both ordered and disordered structures.

## E.2    Screening Based Method

We apply our superconductivity predicting model for screening the entire ICSD database. Potential superconductors are show in Table 10 and 11. To elaborate on the candidates with high confidence, we provide the subsequent details:

1. $CuO_2Sr_{0.075}$ and $Ca_{0.82}CuO_2$ exhibit disordered structures, and their respective parent compounds demonstrate superconductivity [54, 49]. Consequently, these disordered structures are more likely to be superconducting materials as well.

2. $Ba_2CuO_{3.2}$ exhibits superconductivity with a $T_c$ of 70K [28]. Its corresponding parent structure $Ba_2CuO_3$ may also be a superconductor, albeit with a comparatively lower probability.

3. $Ba_{0.83}Fe_2Rb_{0.17}As_2$ and $Ba_{0.6}Fe_2Rb_{0.4}As_2$ share the same parent structure and have closely related compositions. Given that $Ba_{0.6}Fe_2Rb_{0.4}As_2$ exhibits superconductivity with a $T_c$ of 37.5K [42], it is highly likely that $Ba_{0.83}Fe_2Rb_{0.17}As_2$ is also a superconducting material.

Table 12: The novel high-$T_c$ cuprate and h-riched superconducting candidates. Candidates of high confidence are marked in gray.

| Type | Index | Chemical formula | $T_c$ (K) | Reported SC. |
|------|-------|------------------|-----------|--------------|
| Cuprate | 1 | $Ba_2CuCl_2O_2$ | 33.56 | - |
| | 2 | $Tl_2Ca_2Ba_2Cu_3O_{10}$ | 14.09 | - |
| | 3 | $Ba_3CaLa_2GdCu_7O_{17}$ | 10.12 | - |
| | 4 | $YCu_3O_7$ | 9.73 | - |
| | 5 | $BaCaCu_3O_7$ | 9.65 | - |
| | 6 | $Cu_7BO_{16}$ | 7.87 | - |
| | 7 | $CsMgCu_3BiAuO_8$ | 7.82 | - |
| H-riched | 8 | $TbH_3$ | 164.33 | $TbH_3$ 20K [22] Calculated by DFT |
| | 9 | $SeH_3$ | 139.89 | $SeH_3$ 113K [40] Predicted by ML |
| | 10 | $CaGe_2H_9$ | 103.55 | - |
| | 11 | $Ca_2MnCrH_6$ | 58.07 | - |
| | 12 | $SbH_3$ | 46.42 | $SbH_3$ 20K [15] Calculated by DFT |
| | 13 | $MgCoCuH_{42}CS_2N_{16}$ | 44.27 | - |
| | 14 | $Rb_2Ca_2H_4$ | 13.05 | - |

## E.3 Interpretability on SODNet

We attempt to interpret our SODNet predictor by determining which feature(s) a given model weighs most heavily when making the prediction. As shown in Fig. 5, we extract the node embedding of the whole graph in the last layer of SODNet, and present the contributions of each atom to $T_c$ values. We can observe that the B sites contributes more significantly to the property of $T_c$ compared to the Mg site in Fig. 5 (a-d). Moreover, conducting atomic doping and atomic translation on the cation Mg results in a decrease in $T_c$ with 39.0 K $\rightarrow$ 38.4 K $\rightarrow$ 34.3 K. This phenomenon demonstrates that attempting to enhance the $T_c$ value by disrupting the symmetry of Mg site within the lattice may be not workable. Another case of cuprate superconductor has shown in Fig. 5 (e-f), there are three types of oxygen sites that contribute significantly to the $T_c$ value: Hg-O-Hg (PD disorder), Cu-O-Cu (order), and Hg-O-Ba (order). Among them, the contribution of disordered Hg-O-Hg is the greatest, indicating that disrupting the symmetry of oxygen atoms within the lattice might potentially further enhance the property of $T_c$.

## E.4 Generative Superconducting Candidates

We apply our generative model for generating new superconducting candidates. We present the crystal structures of the 20 superconducting candidate materials from Table 6 in Fig. 6. Additionally, we display the 32 superconducting candidate materials in Table 12 and 13, arranged in descending order of predicted $T_c$ values. The structures of all superconducting candidate materials can be obtained in the source code package. We collected superconducting materials that have been reported and observed that five candidates are more likely to be superconducting materials. Among them, four candidates obtained $T_c$ through theoretical calculations, and another material displayed superconducting properties through doping. Specific descriptions are as follows:

1. $SeH_3$ exhibited a $T_c$ of 113K as predicted by machine learning [40], corroborated by DFT calculations indicating 110K [67].

2. DFT methods calculated the $T_c$ of $TbH_3$, $SbH_3$, and $KFe_2As_2$ as 20K [22, 15, 43]. Since H-riched materials belong to conventional superconductors and show high $T_c$ under high pressure, but the conditions for wet experimental synthesis are very stringent. Therefore,

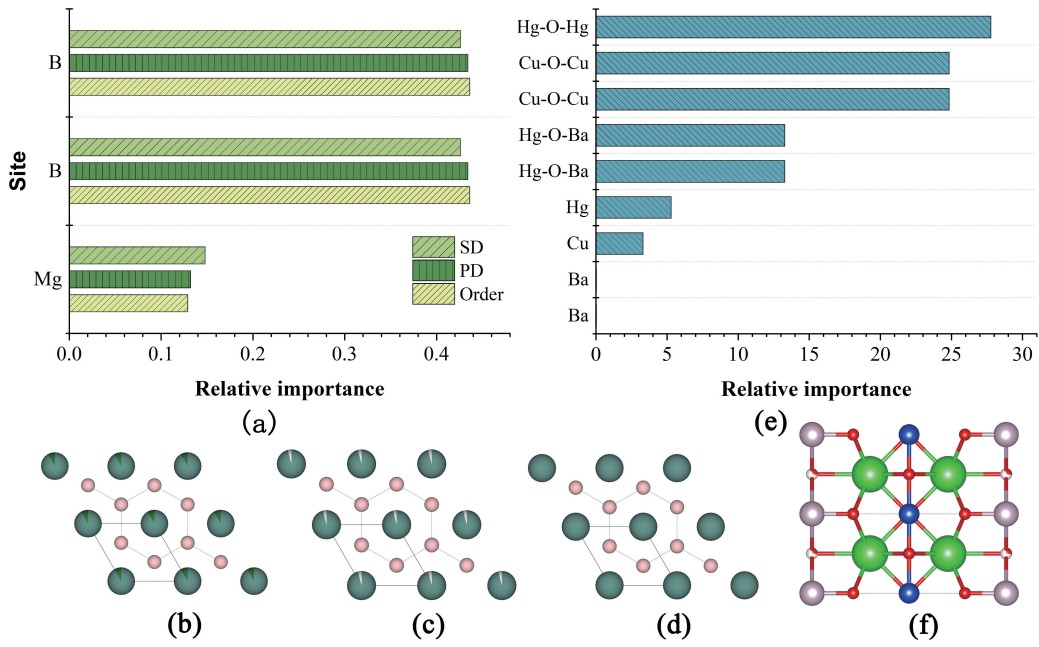

Figure 5: Contribution of each atom to $T_c$ value. (a). Feature relative importance of each site in three type $MgB_2$ superconductors. Snapshots of (b) SD $Mg_{0.9}Al_{0.1}B_2$, (c) PD $Mg_{0.98}B_2$, and (d) ordered $MgB_2$ crystals. Here, B Mg and Al sites are colored by light pink, atrovirens and dark atrovirens. (e). Feature relative importance of $Ba_2CuHgO_{4.27}$ superconductor. (f). Snapshot of $Ba_2CuHgO_{4.27}$ superconductor (Ba: green, Cu: blue, Hg: pink, O: red).

it can further verify whether superconducting materials are superconducting materials by combining DFT methods, and reduce the research and development cycle of superconducting materials.

3. The parent compound SmFeAsO underwent a superconducting $T_c$ around 54 K [1], following fluorine (F) doping at the O-site in the SmO layer. This case can provide us with a method that we can use DiffCSP-SC's generative model to generate superconducting parent structures, and then improve the $T_c$ of materials by doping, or transform materials without superconducting properties into superconducting materials.

Table 13: The novel high-$T_c$ heavy-fermion, iron-based and others superconducting candidates. Candidates of high confidence are marked in gray.

| Type | Index | Chemical formula | $T_c$ (K) | Reported SC. |
|------|-------|------------------|-----------|--------------|
| Heavy-Fermion | 15 | Th | 43.61 | - |
| | 16 | $Ba_3Pu$ | 44.81 | - |
| | 17 | $ThC_3$ | 17.96 | - |
| | 18 | Lu | 4.86 | - |
| | 19 | Yb3In | 1.04 | - |
| Iron-based | 20 | $BaFe_2Se_2$ | 11.99 | - |
| | 21 | SmFeAsO | 4.42 | $SmFeAsO_{0.8}F0.2$ 54K [1] |
| | 22 | $KFe_2As_2$ | 4.23 | $KFe_2As_2$@30GPa 20K [43] Calculated by DFT |
| | 23 | NdFeAsF | 4.13 | - |
| | 24 | FeSe | 3.36 | - |
| Others | 25 | $Ba_3Ca$ | 80.04 | - |
| | 26 | $Ba_2Se$ | 60.70 | - |
| | 27 | Ba | 52.26 | - |
| | 28 | $Mg_3B$ | 43.96 | - |
| | 29 | $BaCl_2O$ | 35.72 | - |
| | 30 | $Ba_2CaB$ | 32.77 | - |
| | 31 | $Sb_2Ba_4$ | 22.70 | - |
| | 32 | $V_3Si_{11}$ | 16.28 | - |

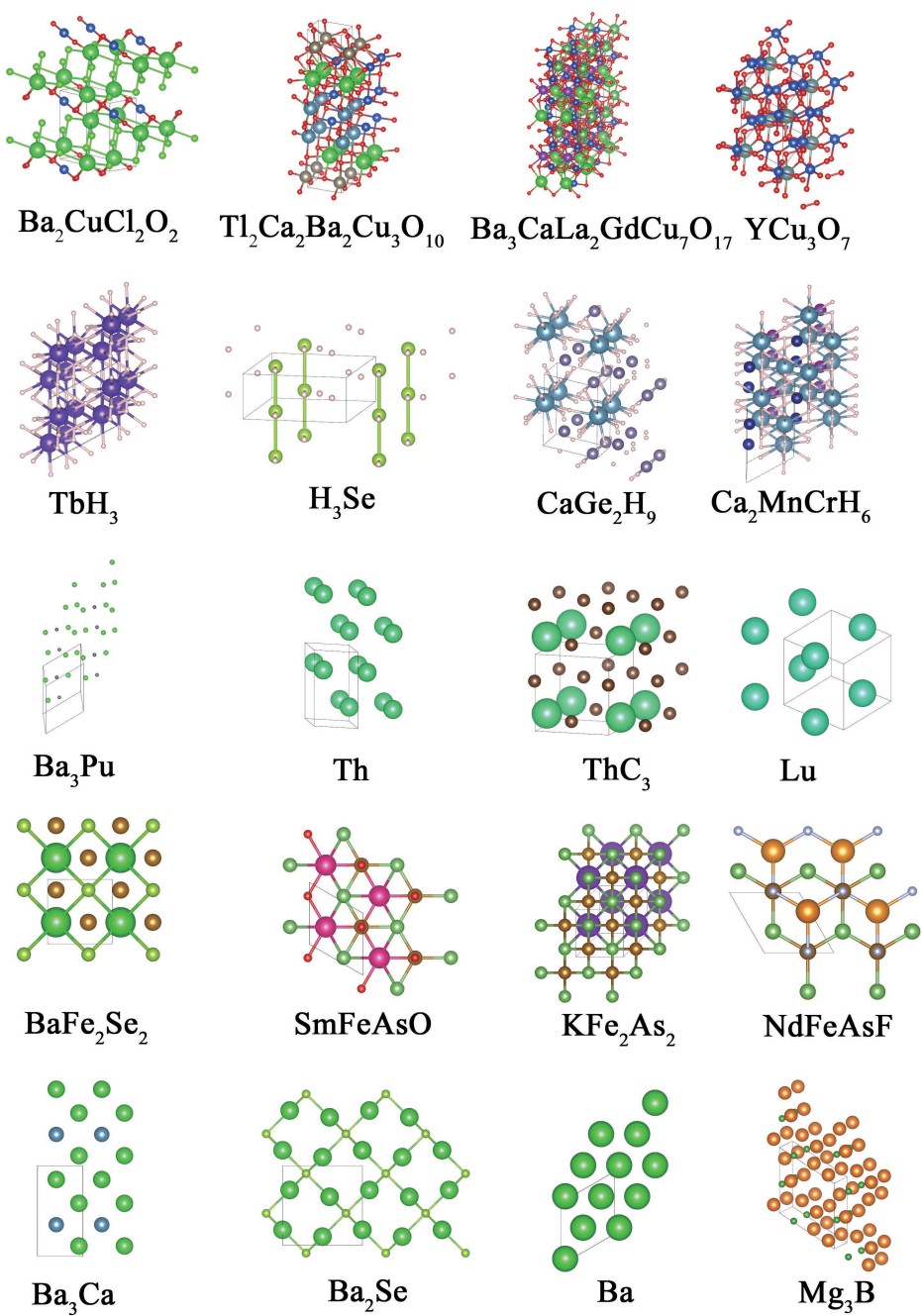

Figure 6: The geometric structures of novel superconducting candidates in Table 6.

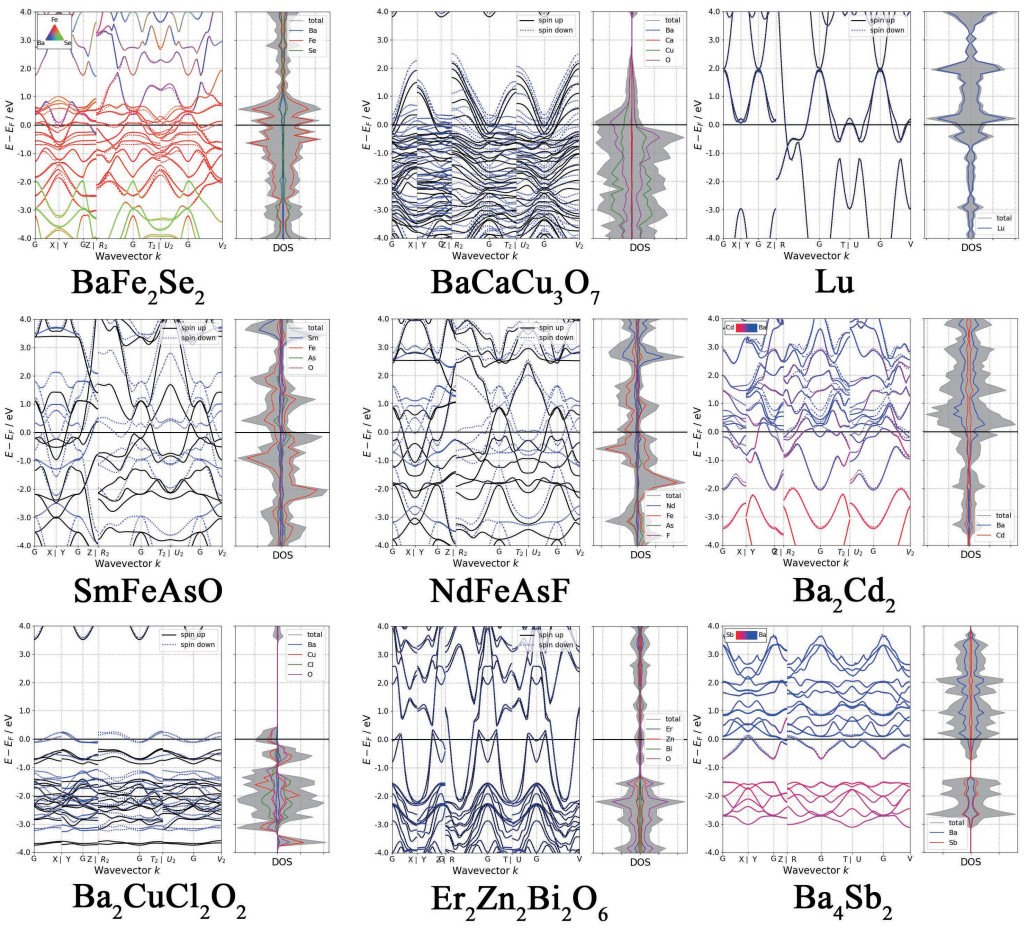

Figure 7: The electronic structures of novel superconducting candidates.

## E.5 DFT Calculations

We conduct DFT calculation using the Vienna ab initio package (VASP) [57, 7]. The structures are fully relaxed using the generalized gradient approximation (GGA) [41] of the SCAN meta-GGA functional, employing the pseudopotentials of the projector augmented wave (PAW) method [4]. A plane wave cutoff of 500 eV is employed for all simulations. Brillouin-zone integrations are performed using the $\tau$-centered Monkhorst-Pack (MP) scheme [37]. We initiate the calculations with a k-point meth featuring a dense sampling density of $2\pi \times 0.04$. The convergence criteria for energy and force is set to 0.1 meV and 0.001 eV/Å, respectively.

The van Hove singularity (VHS) is a notable occurrence in condensed matter physics, specifically in the density of states (DOS) of a material. It manifests as a distinct peak or divergence in the DOS at a particular energy level. We select materials from Table 12 and 13 for DFT calculations and display their band structures and density of states (DOS) in Fig. 7. From the density of states (DOS) plot, we can observe the van Hove singularity (VHS) phenomenon. Additionally, we can also observe the presence of flat bands in the band structures of materials such as $Ba_2CuCl_2O_2$, Lu, $Ba_4Sb_2$, and others. The integration of flat bands in the electronic architecture, along with the Van Hove Singularities (VHS) in the Density of States (DOS), markedly amplifies the likelihood of these candidates being superconducting materials.

# F  Pipeline for Designing Real-world Superconductors.

Fig. 8 presents a pipeline for designing SC., validating our dataset and models for real-world scenarios. We initially generate potential, ordered superconducting structures using the DiffCSP-SC model trained on the SuperCon3D database. Candidate materials are selected based on $T_c$ values predicted by SODNet, followed by DFT verification to confirm the presence of superconducting electronic structures, such as VHS. Subsequently, selected candidates undergo wet lab synthesis, with $T_c$ values characterized and recorded in the SuperCon3D database. Further, if a superconductor is discovered, methods such as doping, which may transform ordered structures into disordered ones, are explored to enhance the $T_c$ value. SODNet is employed to investigate the relationship between disordered structures and doping ratios, aiming to design optimal doping proportions for experimental verification. These experimental outcomes are also recorded in the database. Continuous expansion of the database will incrementally improve the accuracy of the DiffCSP-SC and SODNet models trained on this dataset, creating a reinforcing cycle of enhancement.

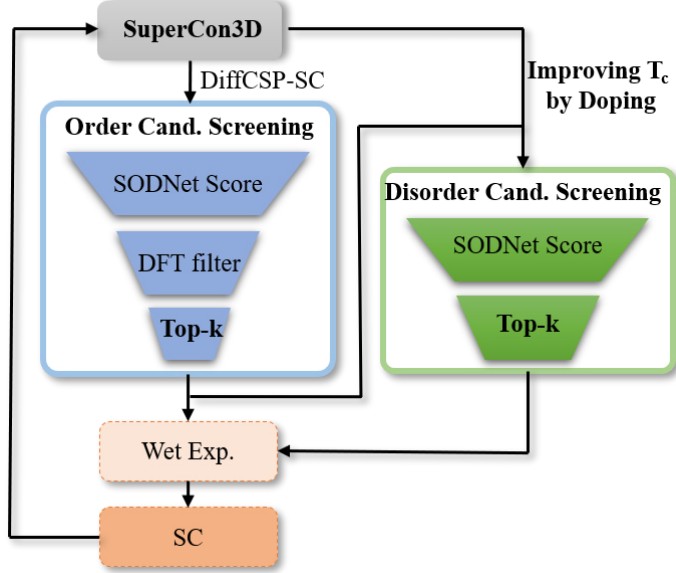

Figure 8: Flowchart for designing novel SC materials.

# G  Code

We have made the source code for SODNet and DiffCSP-SC, as mentioned in this article, available on GitHub. The repositories can be accessed at: https://github.com/pincher-chen/SODNet and https://github.com/pincher-chen/DiffCSP-SC.

