# Supplementary Materials for Learning Superconductivity from Ordered and Disordered Material Structures

# Contents

Submitted to the 38th Conference on Neural Information Processing Systems (NeurIPS 2024) Track on Datasets and Benchmarks. Do not distribute.

The appendix is organized as follows: Section 1 details the collection method and distribution of the SuperCon3D dataset. Section 2 presents more details of disordered graph and models. Section 3 elaborates on the implementation specifics of both property prediction and generative models. The identification of potential superconductors and their corresponding DFT computational outcomes are presented in Section 4. A systematic approach for the design of practical superconductors is expounded in Section 5. We present the limitation of our data and models in Section 6. Section 7 provides the repository link for the associated coding resources.

# 1    SuperCon3D data details

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

Following $S$ message-passing layers, lattice noise $\hat{\epsilon}_L$ is computed as follows:

$$\hat{\epsilon}{L} = {L}\varphi_L\Big(\frac{1}{N}\sum i = 1^N {h}_i^{(S)}\Big),\tag{5}$$

with $\varphi_L$ shaping output as $3 \times 3$. For fractional coordinate score $\hat{\epsilon}_F$, we have:

$$\hat{\epsilon}_{F}[:, i] = \varphi_F({h}_i^{(S)}),\tag{6}$$

where $\hat{\epsilon}_F[:, i]$ is the $i$-th column, and $\varphi_F$ operates on the final layer's output.

The inner product ${L}^\top {L}$ in Eq.(2) ensures O(3)-invariance, as $({Q}{L})^\top ({Q}{L}) = {L}^\top {L}$ for any orthogonal ${Q} \in \mathbb{R}^{3\times 3}$. This guarantees the O(3)-invariance of $\varphi_L$ in Eq.(6), and ${L}$ left-multiplied with $\varphi_L$ ensures O(3)-equivariance of $\hat{\epsilon}_L$. Thus, $\phi({L}, {F}, {A}, t)$ satisfies the proposed properties. More details are described in Jiao et al. [2023].

# 3  Hyper-parameters and training details

In this section, we provide the training details of property predicting models and generative models.

## 3.1  Property predicting models

We employ the codebase from SchNet Schütt et al. [2018][2], CGCNN Xie and Grossman [2018][3], DimNet++ Gasteiger et al. [2020][4], SphereNet Liu et al. [2022][5],ALIGNN Choudhary and DeCost [2021][6], Matformer Yan et al. [2022][7] and MEGNet Chen et al. [2019][8] for baseline implementations. All models are conducted 10-fold experiments based data splited method of 8:1:1. The training details of each model are as follows:

### 3.1.1  SchNet.

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

 Tc of 113K as predicted by machine learning Novakovic et al. [2023], corroborated by DFT calculations indicating 110K Zhang et al. [2015].

2. DFT methods calculated the $T_c$ of $TbH_3$, $SbH_3$, and $KFe_2As_2$ as 20K Hai et al. [2021], Fu et al. [2016], Ptok et al. [2020]. Since H-riched materials belong to conventional superconductors and show high Tc under high pressure, but the conditions for wet experimental synthesis are very stringent. Therefore, it can further verify whether superconducting materials are superconducting materials by combining DFT methods, and reduce the research and development cycle of superconducting materials.

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

## 7  Code

We have provided the source code of SODNet and DiffCSP-SC, which are mentioned in this article, on an anonymous GitHub repository. The access address is as follows: https://anonymous.4open.science/r/SODNet-F569, https://anonymous.4open.science/r/DiffCSP-SC-8F3F.