# OpenReview forum: "Learning Superconductivity from Ordered and Disordered Material Structures"
_NeurIPS.cc/2024/Datasets_and_Benchmarks_Track — NeurIPS 2024 Track Datasets and Benchmarks Poster_

### Official Review · Reviewer_zdb5 · 2024-07-24
**Rewview of SuperCon3D**

**Rating:** 7
**Confidence:** 4
**Correctness:** Yes.
**Clarity:** Yes.

**Review:**

See Strengths and Limitations.

**Strengths:**

1. The paper studies a novel crystal superconductivity problem.
2. The paper proposes a dataset that integrates 3D crystal structures and experimental superconducting transition temperatures.
3. The authors propose two equivariant models.
4. The authors screened the ICSD database to identify potential superconductors.

**Additional Feedback:**

N/A.

**Documentation:**

Yes.

**Ethics:**

N/A.

**Limitations:**

1. According to Appendix 2.1, interstitial disorder (ID) is identified as a potential disorder phenomenon. It would be beneficial to discuss in more detail whether and how the proposed framework can manage ID structures.
2. This paper employs DFT calculations to verify the potential for superconductivity. Providing more details of the DFT settings and pipelines would enhance the reproducibility of the results presented in the paper.

**Opportunities For Improvement:**

See Limitations.

**Relation To Prior Work:**

Yes.

**Summary And Contributions:**

The paper introduces a novel dataset, SuperCon3D, which integrates 3D crystal structures and experimental superconducting transition temperatures. The authors propose two models to showcase the possible methods for exploring the dataset. They present a list of candidate superconductors for future experimental validation.

---

> ### Author Rebuttal · Authors · 2024-08-16
>
> > **L1: According to Appendix 2.1, interstitial disorder (ID) is identified as a potential disorder phenomenon. It would be beneficial to discuss in more detail whether and how the proposed framework can manage ID structures.**
>
> Our model effectively handles ID by carefully considering atomic occupancy and interatomic distances in both node and edge construction.
> For nodes, we use a generalized representation where atomic occupancy \(w_{i,k}\) is fully integrated into the model. The formula:
> $$
> \sum_k w_{i,k} a_{i,k}
> $$
> ensures that all atomic contributions, including those from interstitial sites, are accurately captured.
>
> Edges are constructed based on the radial distribution function (RBF) of interatomic distances:
> $$
> E = w_i w_j RBF(\|\vec{r}_{ij}\|)
> $$
> This approach takes into account the distance between sites, including those involving interstitial atoms, and ensures accurate interaction modeling. The model also incorporates atomic radii to avoid overly strong interactions due to disordered sites.
>
> > **L2: This paper employs DFT calculations to verify the potential for superconductivity. Providing more details of the DFT settings and pipelines would enhance the reproducibility of the results presented in the paper.**
>
> Thank you for the reviewer’s suggestion. We will include additional details in Section 4.5 of the appendix. This will cover specifics such as the DFT software version, exchange-correlation functional, plane wave cutoff energy, k-point mesh, pseudopotential types, SCF convergence criteria, and the steps used to verify superconductivity. We believe these additions will significantly improve the transparency of our study and facilitate the reproducibility of our findings by other researchers.

---

> > ### Author Response · Authors · 2024-08-26
> >
> > Thank you very much for your positive evaluation and constructive feedback on our manuscript. We sincerely appreciate your support and the time you have dedicated to reviewing our work.
> >
> > As the discussion phase is still ongoing, we wanted to reach out to see if you have any additional suggestions or comments that could further enhance the quality of our work. We are open to any further feedback and would be happy to address any remaining concerns you might have.
> >
> > Thank you once again for your valuable insights and contributions.

---

> > > ### Comment · Reviewer_zdb5 · 2024-08-26
> > >
> > > Thanks for the response. My concerns have been addressed.

---

### Official Review · Reviewer_MswQ · 2024-07-25
**A novel dataset for superconductor feature prediction and two models**

**Rating:** 4
**Confidence:** 3
**Correctness:** The dataset is rigorously constructed.
**Clarity:** The clarity of this paper is poor. De…

**Review:**

This paper proposes a novel dataset for superconductor critical temperature prediction. The proposed models perform better than some pervious models on critical temperature prediction. Some new candidate superconductors are generated. However, this paper may be strengthened by doing a more comprehensive comparison with related works, and using a consistent notation.

Strengths:
1. This paper proposes a novel dataset with superconductor 3D structures and their features. This dataset may provide the community with benchmark for superconductor critical temperature prediction.
2. The proposed models for critical temperature prediction and superconductor generation show better performance than some previous models.
3. Some candidate superconductors are generated, which may bring new insights to superconductor studies.

Weakness:
1. The contribution of this paper seems limited, especially in the following aspects:
(i) This paper provides a novel dataset, which is basically generated by combining SuperCon and ICSD. However, a very similar work has been done previously, as in [R1]. This paper has not cited and compare with [R1].
(ii) This paper proposes SODNet for predicting the critical temperature, and compares it with some previous deep-learning based methods. However, in some previous like [R2-3], it is shown that non-deep-learning methods like random forest perform well on this task. For example, [R2] claims to achieve R^2≈0.88, and [R3] for R^2=0.96 (for SVM), far better than 0.748 in this paper. Note that [R2-3] are based on SuperCon, whose samples are similar to SuperCon3D. Yet such non-deep-learning methods are not mentioned by this paper.
(iii) The proposed DiffCSP-SC seems an incrementally improved version of the previous DiffCSP, which only substitutes the backbone from MPNN to transformer. This paper fails to adequately qualify the novelty of this model.
2. The presentation of this paper makes it difficult to understand, such as in the following aspects:
(i) The notation in this paper is inconsistent. For example, α is used in equation (7,8,14), but with three different meanings. Multilayer perceptron is sometimes denoted as φ, and sometimes as ρ.
(ii) ψ is used in equation (16,17) but not defined.
(iii) In section 4.2, only a part of the model DiffCSP-SC is designed, making it hard to understand the whole framework of the model.

[R1] Sommer, Timo, et al. "3DSC-a dataset of superconductors including crystal structures." Scientific Data 10.1 (2023): 816.
[R2] Le, Thanh Dung, et al. "Critical temperature prediction for a superconductor: A variational bayesian neural network approach." IEEE Transactions on Applied Superconductivity 30.4 (2020): 1-5.
[R3] Stanev, Valentin, et al. "Machine learning modeling of superconducting critical temperature." npj Computational Materials 4.1 (2018): 29.

**Strengths:**

1. This paper proposes a novel dataset with superconductor 3D structures and their features. This dataset may provide the community with benchmark for superconductor critical temperature prediction.
2. The proposed models for critical temperature prediction and superconductor generation show better performance than some previous models.
3. Some candidate superconductors are generated, which may bring new insights to superconductor studies.

**Additional Feedback:**

None

**Documentation:**

This paper mentioned how data were collected. The proposed dataset is not publicly available and maintenance is not mentioned. There may be more details in the appendix, but the appendix is not provided at present.

**Ethics:**

No such issues.

**Limitations:**

The limitations have been considered.

**Opportunities For Improvement:**

The authors may consider improvement in the following aspects:
1. Explain how the proposed dataset is different from previous ones like [R1].
2. Explain how SODNet is superior to non-deep-learning methods like those mentioned in [R2-3].
3. Explain the novelty of DiffSCP-SC.

[R1] Sommer, Timo, et al. "3DSC-a dataset of superconductors including crystal structures." Scientific Data 10.1 (2023): 816.
[R2] Le, Thanh Dung, et al. "Critical temperature prediction for a superconductor: A variational bayesian neural network approach." IEEE Transactions on Applied Superconductivity 30.4 (2020): 1-5.
[R3] Stanev, Valentin, et al. "Machine learning modeling of superconducting critical temperature." npj Computational Materials 4.1 (2018): 29.

**Relation To Prior Work:**

This paper has not adequately clarified how it differs from previous works.

**Summary And Contributions:**

This paper provides a novel dataset SuperCon3D which contains the 3D structures and critical temperatures of different superconductors, and proposes a model SODNet for predicting the critical temperature, and a model DiffCSP-SC for superconductor generation. Compared with the broadly used dataset SuperCon, the SuperCon3D dataset adds 3D structures for the materials. The proposed models are tested on SuperCon3D and show better performance than some pervious methods.

---

> ### Author Rebuttal · Authors · 2024-08-16
>
> > **W1: a very similar work has been done previously, as in [R1]. This paper has not cited and compare with [R1].**
>
> Thank you for the thorough review. We recognize the valuable contribution of the 3DSC dataset as described in [R1], and we appreciate the opportunity to clarify the distinctions and contributions of our SuperCon3D dataset.
>
> (1). **Dataset Construction**: As mentioned in the original 3DSC dataset paper (Fig. 3c), approximately 3k structures (partly including non-superconductors) were derived through chemical composition and normalization methods, encompassing most of the parent structures in the SuperCon dataset. An additional 6k structures were generated via artificial doping, primarily producing PD structures. However, real-world materials often exhibit a combination of PD+SD or ID. In contrast, the SuperCon3D dataset is meticulously constructed by rigorously matching experimental data, ensuring both accuracy and representativeness, with a comparable number of parent structures collected.
>
> (2). **Coverage of Hydrogen-Rich Systems**: Hydrogen-rich materials are a key focus in current superconductivity research. The direct matching of SuperCon with ICSD yielded fewer than 10 hydrogen-rich entries. To address this, we incorporated approximately 100 additional entries by collecting the latest literature data, enhancing the dataset’s diversity and relevance in this critical area.
>
> (3). **Manual Matching and Data Verification**: During data collection, we encountered approximately 200 instances where a single chemical formula corresponded to multiple structures. To ensure accuracy, we manually cross-referenced these cases with literature details, including space groups and lattice parameters. Although time-consuming, this process guarantees the transparency and reproducibility of our dataset. We have also provided references doi for both the structures and their corresponding superconducting transition temperatures (Tc) in our code repository.
>
> In the revised manuscript, we will cite and discuss [R1], clearly delineating how SuperCon3D advances the field by addressing limitations in previous work, particularly in handling disordered structures, expanding coverage of hydrogen-rich materials, and ensuring rigorous data validation.
>
> > **W2: ... non-deep-learning methods are not mentioned by this paper.**
>
> Thank you for the suggestion. In response, we conducted additional experiments comparing non-deep learning methods, specifically RF and SVM, using both chemical composition features and combined geometric structure features, following [R1].
>
> | Model                  | R²               | MAE              |
> |------------------------|------------------|------------------|
> | RF-c        | 0.738±0.165      | 0.711±0.050      |
> | SVM-c       | 0.632±0.094      | 0.801±0.041      |
> | RF-geo       | 0.741±0.115      | 0.759±0.051      |
> | SVM-geo      | 0.578±0.114      | 0.827±0.042      |
> | **SODNet**             | **0.748±0.032**  | **0.505±0.055**  |
>
> As shown in the table, the RF model performs similarly to SODNet with chemical composition or geometric structure features. However, SODNet slightly outperforms RF in R² and significantly in MAE, especially with complex structures and disordered systems. We will include a detailed discussion of these performance differences in the revised manuscript.
>
>
> > **W3: The proposed DiffCSP-SC ... fails to adequately qualify the novelty of this model.**
>
> Thank you for the valuable feedback. Our work primarily focuses on designing benchmarks with the SuperCon3D dataset, where we've identified several improvements:
>
> **Transformer Integration**: The original DiffCSP model's use of fully connected edges and redundant MLP calculations was inefficient for the large and complex SuperCon3D dataset. By incorporating Transformer architecture, DiffCSP-SC efficiently aggregates node features and captures essential atomic relationships, boosting performance on complex structures.
>
> **Pretraining + Fine-Tuning**: To overcome the scarcity of superconducting materials, we adopted a "pretraining + fine-tuning" strategy. Large-scale pretraining on over a million crystal structures improved model generalization and significantly enhanced its generative capabilities.
>
> **Optimized Generative Process**: DiffCSP-SC uses a SODNet-based predictor to focus on high-Tc superconductors, improving accuracy and effectiveness.
>
> In summary, as a dataset and benchmark paper, our main contribution is detailing the SuperCon3D dataset, with a focus on disordered structures. DiffCSP-SC, combined with SODNet, also provides a method for generating and evaluating these structures.
>
> > **W4: (i) The notation in this paper is inconsistent.(ii) ψ is used in equation (16,17) but not defined...(iii)only a part of the model DiffCSP-SC is designed.**
>
> Thank you for the feedback. We acknowledge that inconsistent notation and incomplete descriptions impacted clarity. We'll make the following improvements:
>
> (i). We will standardize the notation by using α in equation (7), β in equation (8), and θ in equation (14). Additionally, MLP will consistently be denoted as φ throughout the manuscript.
>
> (ii). ψ executes Fourier Transformation on relative fractional coordinates, ensuring periodic translation invariance.
>
>  (iii). Section 4.2 currently presents only part of the DiffCSP-SC model, which may lead to difficulties in understanding the entire framework. We will expand this section to include a complete description of the model, covering all components and processes.
>
> > **W5: The proposed dataset is not publicly available and maintenance is not mentioned.**
>
> Thank you for your feedback. We've uploaded the appendix, including dataset and code access, to the OpenReview system. The dataset is included in the SODNet source code package.
>
> Thank you again for your valuable suggestions. We believe we have addressed your concerns and kindly hope for your feedback and reconsideration of the scores.

---

> > ### Author Response · Authors · 2024-08-26
> > **Looking forward to your feedback**
> >
> > Dear Reviewer MswQ, as the deadline for the author-reviewer discussion approaches, we kindly request your feedback on whether our responses have satisfactorily addressed your primary concerns. Should you have any additional suggestions or comments, please do not hesitate to share them with us. We would be more than willing to engage in further discussions and make any necessary improvements.
> >
> > Thank you once again for dedicating your valuable time to reviewing our work.

---

### Official Review · Reviewer_ZXok · 2024-07-27
**A benchmark dataset for ordered and disordered superconductors**

**Rating:** 7
**Confidence:** 5
**Correctness:** The claims in the submission are corr…
**Clarity:** Paper is well written.

**Review:**

Overall, the curated datasets have high quality. The 2 benchmark experiments are well designed and the author also provides detailed ablation studies. The paper is well-written and includes a lot of details in the supplementary info. My main concern is that I couldn’t find a link to the dataset and some documentation of the data curation process can be improved. I also have minor concerns regarding some metrics used to evaluate the models. Detailed feedback is provided in the following sections.

**Strengths:**

- The curated dataset includes both experimental structures and experimental Tc values. It is also carefully examined by domain experts.
- The dataset includes disordered structures and categorized several different types of disordered structures. It is an important step forward for the field given the recent discussion around disordered structures from the A-lab paper.
- Detailed benchmark and ablation study is provided for both property prediction models and inverse design models.

**Additional Feedback:**

No additional feedback

**Documentation:**

I couldn’t find a link to the dataset. Are they authors going to release the data? Documentation around the curation process of the data can also be improved.

**Ethics:**

No ethical concerns that warrant further discussion.

**Limitations:**

There is no discussion of the limitation in the paper. I listed a few potential limitations in the previous section. I also suggest the authors discuss the limitations and potential pitfalls in their dataset curation process.

**Opportunities For Improvement:**

- The size of the dataset is relatively small, including only 1578 materials. It is only a small subset of SuperCon database. Why is it difficult to find experimental structures for the rest in SuperCon?
- In the definition of success rate (SR) for inverse design, the authors didn’t discuss whether they are generating novel material or those already existing in training data. Only novel material should be counted as a success in my view.
- The authors use property predictors trained on the same dataset to evaluate their inverse design model. There is a high risk of false positives. I suggest the authors discuss the limitation and propose potential methods to address it.

**Relation To Prior Work:**

Paper has clearly discussed the difference against previous contributions. The difference is significant enough in my view.

**Summary And Contributions:**

This paper presents a dataset including the ordered and disordered 3D structure of superconductors and their Tc values curated from literature. The authors also designed two benchmarks for superconductor discovery: superconductivity prediction and inverse crystal structure generation. They developed new models for each task and achieved SOTA performance in both tasks they defined.

---

> ### Author Rebuttal · Authors · 2024-08-16
>
> > **Q1: Why is it difficult to find experimental structures for the rest in SuperCon?**
>
> Thank you for the question. The limited dataset size is due to the following:
>
> (1). **Data Availability**: SuperCon mainly provides chemical formulas and Tc, but lacks 3D structures. Matching these requires cross-referencing with other databases like ICSD, where many materials, especially disordered ones, have incomplete structural data.
>
> (2). **Scarcity of Experimental Data**: Many superconductors, particularly new or complex ones, lack available 3D structure data, making them difficult to include.
>
> (3). **Complexity of Disordered Structures**: Many materials are derived from a few parent structures, leading to disordered variations that are often underrepresented in databases. Specifically, while SD disorder can be achieved through doping, the generation of PD, ID, and various combinations thereof makes the structure generation challenging.
>
> The SuperCon3D dataset includes most of the parent structures from the SuperCon dataset, along with a small number of structures collected from the literature. We have also provided references DOI for the Tc values and the structures for verification. The dataset is accessible at: https://anonymous.4open.science/r/SODNet-F569.
>
> > **Q2：... the authors didn’t discuss whether they are generating novel material or those already existing in training data.**
>
> Thank you for the detailed review. In response to your suggestion, we have introduced a new metric, the "Novelty Success Rate" (NSR), to specifically quantify the proportion of novel structures generated by the model.
>
> The NSR is defined as:
>
> $$
> NSR_\alpha(\tilde{D}) = \frac{\|\|\tilde{M} \mid \tilde{M} \in \tilde{D}, \varphi(\tilde{M}) > P_{100-\alpha}(D_{train}), \tilde{M} \notin D_{train}\|\|}{\|\|\tilde{D}\|\|}
> $$
>
>
> This metric focuses on evaluating the model's ability to generate structures that are not present in the training dataset.
>
> We conducted additional experiments using NSR, and the results are summarized in the table below, comparing different models:
>
> | Model         | Data               | NSR10 | NSR30 | NSR50 |
> |---------------|--------------------|-------|-------|-------|
> | CDVAE         | O                  | 0.02  | 0.02  | 0.02  |
> | SyMat         | O                  | 0.02  | 0.03  | 0.03  |
> | DiffCSP       | O                  | 0.03  | 0.04  | 0.04  |
> | DiffCSP-SC    | O                  | 0.04  | 0.04  | 0.09  |
> | CDVAE         | Pre-training + O   | 0.19  | 0.19  | 0.25  |
> | SyMat         | Pre-training + O   | 0.20  | 0.21  | 0.26  |
> | DiffCSP       | Pre-training + O   | 0.25  | 0.25  | 0.33  |
> | DiffCSP-SC    | Pre-training + O   | **0.31**  | **0.31**  | **0.39**  |
>
> As shown, our DiffCSP-SC model outperforms others in generating novel materials, as indicated by higher NSR values across all metrics (NSR10, NSR30, NSR50).
>
> Furthermore, our training dataset includes approximately 1 million material structures, many of which have not been experimentally validated for superconductivity. Even if some generated structures appear in the training data, they may still hold potential superconducting properties, making them valuable for further investigation. By leveraging this large dataset and pre-training strategies, our model demonstrates  advantages in generating novel and potentially superconductive structures.
>
> > **Q3: The authors use property predictors trained on the same dataset to evaluate their inverse design model. There is a high risk of false positives. I suggest the authors discuss the limitation and propose potential methods to address it.**
>
> Thank you for the insightful suggestion. We acknowledge the concern regarding potential false positives when using property predictors trained on the same dataset as the inverse design model. This overlap may lead to overfitting, potentially inflating the perceived performance of the model. We will address this concern in the appendix (Section 6: Limitations) with a more detailed discussion and potential solutions. Proposed Solutions including:
>
> (1).**Cross-Validation and Independent Test Sets**: Implementing cross-validation and using independent test sets to evaluate the generative model can help mitigate the risk of overfitting.
>
> (2).**Ensemble of Models**: Using an ensemble of different property prediction models can reduce bias and increase the reliability of the predictions.

---

> > ### Comment · Reviewer_ZXok · 2024-08-17
> > **Thanks for your reponse**
> >
> > I thank the authors for their response. They've properly addressed my remaining concerns. I think this is a good contribution and will keep my original score.

---

> > > ### Author Response · Authors · 2024-08-26
> > >
> > > Thank you very much for your positive feedback and for confirming that our responses have adequately addressed your concerns. We appreciate your time and effort in reviewing our manuscript and your valuable insights that have contributed to improving the quality of our work. We are glad to hear that you consider our contribution to be of value.
> > >
> > > Thank you once again for your support and consideration.

---

### Decision · Program_Chairs · 2024-09-26

**Decision:**

Accept (Poster)

**Comment:**

The paper addresses a very important real world problem by i) proposing a new dataset ii) benchmark as well as iii) a new method. In particular, the dataset is very well curated and has high quality, including expert annotations. All reviewers agree that the dataset is a very valuable contribution, addressing the important real world problem of super conductivity prediction. The concerns of reviewer (MswQ) who is the only one recommending rejection focus mainly on notations for 4 equations and on clarity on the dataset extraction from multiple sources, as well as a more elaborate discussion of the limitations. Some of these concerns had been shared by reviewer  ( ZXok ) but could be clarified relatively easily, e.g. by adding the promised detailed discussion of the limitations. Thus, a more elaborate discussion of the limitations seems to be a point, which can be easily addressed in the camera ready version and does not neglect the effort and valuable contribution of a manual dataset curation for an important problem. We encourage the authors to include some of the discussions with the reviewers in the publication, and in particular include the link to the dataset.
The final point to highlight is that one reviewer ( ZXok ) who engaged in the rebuttal is an expert in the field (by his own and my assessment) and his review and confidence should correspondingly be weighted. This is a valuable paper for the community to further improve and tackle the important problem of super-conductivity prediction and should thus be accepted.